# Weakly and Nearly Countably Compactness in Generalized Topology

Zuhier Altawallbeh [1,*], Ahmad Badarneh [2], Ibrahim Jawarneh [2] and Emad Az-Zo'bi [3]

1 Department of Mathematics, Tafila Technical University, Tafila 66110, Jordan
2 Department of Mathematics, Al-Hussein Bin Talal University, Ma'an 71111, Jordan
3 Department of Mathematics and Statistics, Mutah University, Alkarak 61710, Jordan
* Correspondence: zuhier1980@gmail.com or zuhier@ttu.edu.jo

**Abstract:** We define the notions of weakly $\mu$-countably compactness and nearly $\mu$-countably compactness denoted by $\mathcal{W}\mu$-CC and $\mathcal{N}\mu$-CC as generalizations of $\mu$-compact spaces in the sense of Csaśzaŕ generalized topological spaces. To obtain a more general setting, we define $\mathcal{W}\mu$-CC and $\mathcal{N}\mu$-CC via hereditary classes. Using $\mu_\theta$-open sets, $\mu$-regular open sets, and $\mu$-regular spaces, many results and characterizations have been presented. Moreover, we use the properties of functions to investigate the effects of some types of continuities on $\mathcal{W}\mu$-CC and $\mathcal{N}\mu$-CC. Finally, we define soft $\mathcal{W}\mu$-CC and $\mathcal{N}\mu$-CC as generalizations of soft $\mu$-compactness in soft generalized topological spaces.

**Keywords:** $\mu$-countably compact; $\mu\mathcal{H}$-countably compact; weakly $\mu$-countably compact; nearly $\mu$-countably compact





## 1. Introduction

In 2002, Csaśzaŕ introduced generalized topology [1]. Csaśzaŕ's topology removes the intersection property of a finite number of open sets. Many authors have made different generalizations of compactness such as [2–5]. On the other hand, many generalizations have been done by using the notion of generalized topology as [6–10]. In particular, we introduce the notion of weakly (nearly) $\mu$-countably compactness. Additionally, by using hereditary classes defined in 2007 [8], weakly (nearly) $\mu\mathcal{H}$-countably compact spaces have been investigated in more general settings. The current paper has an application in soft set theory as can be seen in the last section. Similar applications can be made in fuzzy and set theories, which are in uncertainty in mathematics. In particular, many developments can be made as interactions between uncertainty and other disciplines of mathematics as fractional calculus or in function spaces. So, the reader can return to [11–15].

A subset $\mu$ of the power set of $X$ is generalized topology on $X$, whenever $\phi \in \mu$ and $\bigcup_{\alpha \in \Delta} A_\alpha \in \mu$ for all $A_\alpha \in \mu$ [8]. In this work, the notation $\mu$ stands for strong generalized topology, which means $X \in \mu$. A subset A is $\mu$-open whenever $A \in \mu$ and $A$ is $\mu$-closed if $X \backslash A \in \mu$. The interior of $A$ in $\mu$ is $Int_\mu(A) = \bigcup_{S_\alpha \subseteq A} S_\alpha$ for all $S_\alpha \in \mu$, and the closure is given by $Cl_\mu(A) = \bigcap_{A \subseteq F_\alpha} F_\alpha$ for all $X \backslash F_\alpha \in \mu$. Whenever $A = Int_\mu(Cl_\mu(A))$ (resp. $A = Cl_\mu(Int_\mu(A))$), then A is called $\mu$-regular open (resp. $\mu$-regular closed) [8]. See that whenever $A = Int_\mu(A)$, then A is $\mu$-open [6]. We write the pair $(X, \mu)$ simply as $X_\mu$. Now, let $A \neq \emptyset$ be a subset of $X_\mu$, then $\mu_A$ is a generalized subspace topology of $A$ in $X$ whenever, for all $B \in \mu_A$, there is a subset $U \in \mu$ such that $B = U \cap A$ [16]. Let $\mathcal{H} \subseteq \mathcal{P}(X)$ and $\emptyset \in \mathcal{H}$, then $\mathcal{H}$ is a hereditary class on $X$ whenever $C \in \mathcal{H}$ and $A \subseteq C$, then $A \in \mathcal{H}$ for all $A, C \subseteq X$. The pair $(X_\mu, \mathcal{H})$ is a generalized space with respect to $\mathcal{H}$ [8]. Moreover, whenever $A \cup B \in \mathcal{H}$ for all $A, B \in \mathcal{H}$, then $\mathcal{H}$ is called an ideal on $X$.

Next, we give basic concepts of known generalizations of compactness and countable compactness in generalized topology. Nearly $\mu$-countably compactness and $\mu\mathcal{H}$-countably

compactness have been discussed in Section 2. In $\mu$-regular spaces, Theorem 4 shows that there is no difference between nearly $\mu\mathcal{H}$-countably compact space and $\mu\mathcal{H}$-countably compact space. In Section 3, weakly $\mu$-countably compactness has been characterized by using $\mu$-closed sets in Theorem 10. There have been some further results about subsets of weakly $\mu$-countably compact spaces. Some examples are given to verify the new spaces. The main contribution in Section 4 is to characterize the continuity in the generalized topology of the discussed spaces. Theorems 23 and 24 show that continuity preserves such given spaces. Using different kinds of continuity, we obtain stronger results in several theorems in Section 4. As a consequence, we add Section 5 before the conclusions. The short section is about an applicable definition in soft theory that generalizes soft $\mu$-compactness.

**Definition 1** ([7]). *Let $X$ be a set. The space $X_\mu$ is said to be $\mu$-compact whenever $X = \bigcup\limits_{\lambda \in \Lambda} U_\lambda$, where $U_\lambda \in \mu$ for all $\lambda \in \Lambda$, then there is a finite sub-collection $\{U_\lambda : \lambda \in \Lambda_0 \subseteq \Lambda\}$ such that $X = \bigcup\limits_{\lambda \in \Lambda_0} U_\lambda$.*

**Definition 2** ([17]). *Let $X$ be a set. The space $X_\mu$ is said to be nearly $\mu$-compact (denoted by $\mathcal{N}\mu$-compact) whenever $X = \bigcup\limits_{\lambda \in \Lambda} U_\lambda$, where $U_\lambda \in \mu$ for all $\lambda \in \Lambda$, then there is a finite sub-collection $\{U_\lambda : \lambda \in \Lambda_0 \subseteq \Lambda\}$ such that $X = \bigcup\limits_{\lambda \in \Lambda_0} Int_\mu Cl_\mu(U_\lambda)$.*

**Definition 3** ([10]). *Let $X$ be a set. The space $X_\mu$ is said to be weakly $\mu$-compact (denoted by $\mathcal{W}\mu$-compact) whenever $X = \bigcup\limits_{\lambda \in \Lambda} U_\lambda$, where $U_\lambda \in \mu$ for all $\lambda \in \Lambda$, then there is finite sub-collection $\{U_\lambda : \lambda \in \Lambda_0 \subseteq \Lambda\}$ such that $X = \bigcup\limits_{\lambda \in \Lambda_0} Cl_\mu(U_\lambda)$.*

**Definition 4** ([18]). *Let $(X_\mu, \mathcal{H})$ be a space with respect to $\mathcal{H}$. The pair $(X_\mu, \mathcal{H})$ is said to be weakly $\mu\mathcal{H}$-compact (denoted by $\mathcal{W}\mu\mathcal{H}$-compact) whenever $X = \bigcup\limits_{\lambda \in \Lambda} U_\lambda$, where $U_\lambda \in \mu$ for all $\lambda \in \Lambda$, then there is a finite sub-collection $\{U_\lambda : \lambda \in \Lambda_0 \subseteq \Lambda\}$ such that $X \backslash \bigcup\limits_{\lambda \in \Lambda_0} Cl_\mu(U_\lambda) \in \mathcal{H}$.*

**Definition 5** ([17]). *Let $(X_\mu, \mathcal{H})$ be a space with respect to $\mathcal{H}$. The pair $(X_\mu, \mathcal{H})$ is said to be nearly $\mu\mathcal{H}$-compact (denoted by $\mathcal{N}\mu\mathcal{H}$-compact) whenever $X = \bigcup\limits_{\lambda \in \Lambda} U_\lambda$, where $U_\lambda \in \mu$ for all $\lambda \in \Lambda$, then there is a finite sub-collection $\{U_\lambda : \lambda \in \Lambda_0 \subseteq \Lambda\}$ such that $X \backslash \bigcup\limits_{\lambda \in \Lambda_0} Int_\mu Cl_\mu(U_\lambda) \in \mathcal{H}$.*

**Definition 6** ([19]). *Let $X$ be a set. The space $X_\mu$ is said to be $\mu$-countably compact (denoted by $\mu$-CC) whenever $X = \bigcup\limits_{\lambda \in \Lambda} U_\lambda$, where $U_\lambda \in \mu$ for all $\lambda \in \Lambda$ and $\Lambda$ is a countable index set, then there is a finite sub-collection $\{U_\lambda : \lambda \in \Lambda_0 \subseteq \Lambda\}$ such that $X = \bigcup\limits_{\lambda \in \Lambda_0} U_\lambda$.*

**Definition 7** ([19]). *Let $X_\mu$ be a space. A subset $A$ of $X$ is said to be $\mu$-CC set whenever $A \subset \bigcup\limits_{\lambda \in \Lambda} U_\lambda$, where $U_\lambda \in \mu$ for all $\lambda \in \Lambda$ and $\Lambda$ is a countable index set, then there is a finite sub-collection $\{U_\lambda : \lambda \in \Lambda_0 \subseteq \Lambda\}$ such that $A \subset \bigcup\limits_{\lambda \in \Lambda_0} (U_\lambda)$.*

**Definition 8** ([19]). *Let $(X_\mu, \mathcal{H})$ be a space with respect to $\mathcal{H}$. The pair $(X_\mu, \mathcal{H})$ is said to be $\mu\mathcal{H}$-countably compact (denoted by $\mu\mathcal{H}$-CC) whenever $X = \bigcup\limits_{\lambda \in \Lambda} U_\lambda$, where $U_\lambda \in \mu$ for all $\lambda \in \Lambda$ and $\Lambda$ is a countable index set, then there is a finite sub-collection $\{U_\lambda : \lambda \in \Lambda_0 \subseteq \Lambda\}$ such that $X \backslash \bigcup\limits_{\lambda \in \Lambda_0} U_\lambda \in \mathcal{H}$.*

**Definition 9** ([10]). *Let $X$ be a set. The space $X_\mu$ is said to be $\mu$-regular whenever, for each $\mu$-open subset $U$ of $X$ and for each $x \in U$, there exist a $\mu$-open subset $V$ of $X$ and a $\mu$-closed subset $F$ of $X$ such that $x \in V \subset F \subset U$.*

**Definition 10** ([10]). *If $C \subseteq X_\mu$ and $x \in X$, then $x$ is called $\theta_\mu$-cluster point of $C$ if $Cl_\mu(V) \cap C \neq \varnothing$ for all $V \in \mu$ and $x \in V$. The set $(Cl_\mu)_\theta(C) = \{x \in X : x$ is a $\theta_\mu$-cluster point of $C\}$ if $(Cl_\mu)_\theta(C) = C$, then $C$ is called $\mu_\theta$-closed. The set $C$ is $\mu_\theta$-open if $X \backslash C$ is $\mu_\theta$-closed.*

**Lemma 1** ([10]). *If $A, C \subseteq X_\mu$ and $A \subseteq C$, then $Cl_{\mu_C}(A) = Cl_\mu(A) \cap C$.*

**Lemma 2** ([10]). *Let $f : X_\mu \to Y_\beta$ be a function. The following statements are equivalent:*

1. *$f$ is $(\mu, \beta)$-continuous;*
2. *$f(Cl_\mu(U)) \subset Cl_\beta(f(U))$, for all $U \subseteq X$;*
3. *$Cl_\mu f^{-1}(V) \subset f^{-1}(Cl_\beta(V))$, for all $V \subseteq Y$.*

**Definition 11.** *Let $f : X_\mu \to Y_\beta$ be a function. If for each $t \in X$ and $f(t) \in V \in \beta$, there exists $U \in \mu$ containing $t$ such that:*

1. *$f(Cl_\mu(U)) \subseteq V$, then $f$ is said to be strongly $\varnothing(\mu, \beta)$-continuous [20].*
2. *$f(Int_\mu Cl_\mu(U)) \subseteq V$, then $f$ is said to be super $(\mu, \beta)$-continuous [20].*
3. *$f(Int_\mu Cl_\mu(U)) \subseteq Int_\beta Cl_\beta(V)$, then $f$ is said to be $(\delta, \delta')$-continuous [21].*
4. *$f(U) \subseteq Int_\beta Cl_\beta(V))$, then $f$ is said to be almost $(\mu, \beta)$-continuous [22].*

## 2. Nearly $\mu$-Countably Compactness and Nearly $\mu\mathcal{H}$-Countably Compactness

　　In this section, we introduce the notion of nearly $\mu$-countably compact and the notion of nearly $\mu\mathcal{H}$-countably compact. Some interesting examples are presented to investigate these spaces.

**Definition 12.** *Let $X$ be a set. The space $X_\mu$ is said to be nearly $\mu$-countably compact (denoted by $\mathcal{N}\mu$-CC) whenever $X = \bigcup\limits_{\lambda \in \Lambda} U_\lambda$, where $U_\lambda \in \mu$ for all $\lambda \in \Lambda$ and $\Lambda$ is a countable index set, then there is a finite sub-collection $\{U_\lambda : \lambda \in \Lambda_0 \subseteq \Lambda\}$ such that $X = \bigcup\limits_{\lambda \in \Lambda_0} Int_\mu Cl_\mu(U_\lambda)$.*

**Corollary 1.** *Every $\mu$-CC space is $\mathcal{N}\mu$-CC space.*

**Proof.** Let $X_\mu$ be a $\mu$-CC space. Which means that $X = \bigcup\limits_{\lambda \in \Lambda} U_\lambda$, where $U_\lambda \in \mu$ for all $\lambda \in \Lambda$ and $\Lambda$ is a countable index set, then there is a finite sub-collection $\{U_\lambda : \lambda \in \Lambda_0 \subseteq \Lambda\}$ where $X = \bigcup\limits_{\lambda \in \Lambda_0} U_\lambda$, but $U_\lambda \subseteq Int_\mu Cl_\mu(U_\lambda)$ for each $\lambda \in \Lambda_0$, so $\bigcup\limits_{\lambda \in \Lambda_0} (U_\lambda) \subseteq \bigcup\limits_{\lambda \in \Lambda_0} Int_\mu Cl_\mu(U_\lambda)$. Thus, $X = \bigcup\limits_{\lambda \in \Lambda_0} Int_\mu Cl_\mu(U_\lambda)$. □

　　The converse of Corollary 1 is not true as presented in Example 1.

**Example 1.** *Let $(\mathbb{R}, \mu)$ be a space, where $\mu = \{A \subseteq \mathbb{R} : A = \varnothing$ or $\mathbb{R} \backslash A$ is a countable$\}$. Let $\mathbb{R} = \bigcup\limits_{\lambda \in \Lambda} U_\lambda$, where $U_\lambda \in \mu$ for all $\lambda \in \Lambda$ and $\Lambda$ is a countable index set, then we can find a finite sub-collection $\{U_\lambda : \lambda \in \Lambda_0 \subseteq \Lambda\}$, so $Cl_\mu(U_\lambda) = \mathbb{R}$ and $Int_\mu Cl_\mu(U_\lambda) = \mathbb{R}$ for each $\lambda \in \Lambda_0$. Thus $\mathbb{R} = \bigcup\limits_{\lambda \in \Lambda_0} Int_\mu Cl_\mu(U_\lambda)$ is a $\mathcal{N}\mu$-CC space. It is clear that $(\mathbb{R}, \mu)$ is not $\mu$-CC space.*

**Definition 13.** *Let $(X_\mu, \mathcal{H})$ be a space with respect to $\mathcal{H}$. The pair $(X_\mu, \mathcal{H})$ is said to be nearly $\mu\mathcal{H}$-countably compact (denoted by $\mathcal{N}\mu\mathcal{H}$- CC) whenever $X = \bigcup\limits_{\lambda \in \Lambda} U_\lambda$, where $U_\lambda \in \mu$ for all $\lambda \in \Lambda$ and $\Lambda$ is a countable index set, then there is a finite sub-collection $\{U_\lambda : \lambda \in \Lambda_0 \subseteq \Lambda\}$ such that $X \backslash \bigcup\limits_{\lambda \in \Lambda_0} Int_\mu Cl_\mu(U_\lambda) \in \mathcal{H}$.*

**Theorem 1.** *If $X$ is a $\mathcal{N}\mu$-CC space, then $X$ is a $\mathcal{N}\mu\mathcal{H}$-CC space.*

**Proof.** Let $X$ be a $\mathcal{N}\mu$-CC space. Which means that $X = \bigcup_{\lambda \in \Lambda} U_\lambda$, where $U_\lambda \in \mu$ for all $\lambda \in \Lambda$ and $\Lambda$ is a countable index set, then there is a finite sub-collection $\{U_\lambda : \lambda \in \Lambda_0 \subseteq \Lambda\}$ where $X = \bigcup_{\lambda \in \Lambda_0} Int_\mu Cl_\mu(U_\lambda)$, but $X \backslash \bigcup_{\lambda \in \Lambda_0} Int_\mu Cl_\mu(U_\lambda) = \varnothing \in \mathcal{H}$. Hence, $X_\mu$ be $\mathcal{N}\mu\mathcal{H}$-CC space. $\square$

In Example 1, we show that the converse of Theorem 1 is not always true.

**Example 2.** *Let $X = \mathbb{Z}$, and $\mathcal{B} = \{\{2n-1, 2n, 2n+1\} : n \in \mathbb{Z}\}$ be $\mu$-subbase where $\mu$ generated by $\mathcal{B}$ such that $(X, \mu(\mathcal{B}))$ and $\mathcal{H} = \mathcal{P}(\mathbb{Z})$. Then, $(X, \mu(\mathcal{B}))$ is not $\mathcal{N}\mu$-CC space. However, it is $\mathcal{N}\mu\mathcal{H}$-CC space. Since $X = \bigcup_{\lambda \in \Lambda} U_\lambda$, where $U_\lambda \in \mu$ for all $\lambda \in \Lambda$ and $\Lambda$ is a countable index set, then there is a finite sub-collection $\{U_\lambda : \lambda \in \Lambda_0 \subseteq \Lambda\}$ where $X \backslash \bigcup_{\lambda \in \Lambda_0} Int_\mu Cl_\mu(U_\lambda) \in \mathcal{H}$.*

**Theorem 2.** *If $X$ is a $\mu\mathcal{H}$-CC space, then $X$ is a $\mathcal{N}\mu\mathcal{H}$-CC space.*

**Proof.** Let $X$ be a $\mu\mathcal{H}$-CC space. This means for $X = \bigcup_{\lambda \in \Lambda} U_\lambda$, where $U_\lambda \in \mu$ for all $\lambda \in \Lambda$ and $\Lambda$ is a countable index set, then there is a finite sub-collection $\{U_\lambda : \lambda \in \Lambda_0 \subseteq \Lambda\}$ where $X \backslash \bigcup_{\lambda \in \Lambda_0} U_\lambda \in \mathcal{H}$, but $X \backslash \bigcup_{\lambda \in \Lambda_0} Int_\mu Cl_\mu(U_\lambda) \subseteq X \backslash \bigcup_{\lambda \in \Lambda_0} (U_\lambda)$. Thus, $X \backslash \bigcup_{\lambda \in \Lambda_0} Int_\mu Cl_\mu(U_\lambda) \in \mathcal{H}$. Hence, $X_\mu$ is a $\mathcal{N}\mu\mathcal{H}$-CC space. $\square$

The converse of Theorem 2 is not true, as presented in Example 3.

**Example 3.** *Let $X = (0, 1)$, $\mu = \{\phi, G_n : n \in \mathbb{Z}^+\}$, where $G_n = (\frac{1}{n}, 1)$ and $\mathcal{H} = \mathcal{H}_f$. Then, $X_\mu$ is $\mathcal{N}\mu\mathcal{H}$-CC because for any proper $\mu$-open set $Int_\mu Cl_\mu(G_{n_i}) = X$ where $i \in \mathbb{Z}^+$, then $X \backslash \bigcup_{i}^{n} Int_\mu Cl_\mu(G_{n_i}) \in \mathcal{H}$. However, that is not $\mu\mathcal{H}$-CC because there is no finite sub-collection such that $X \backslash \bigcup_{k}^{n} G_{n_i} \in \mathcal{H}$.*

**Theorem 3.** *If a space $X_\mu$ is $\mathcal{N}\mu\mathcal{H}$-CC, then for every countable cover of $X$ by $\mu_\theta$-open sets, there exists a finite sub-collection $\{U_\lambda : \lambda \in \Lambda_0 \subseteq \Lambda\}$ such that $X \backslash \bigcup_{\lambda \in \Lambda_0} U_\lambda \in \mathcal{H}$.*

**Proof.** Suppose $(X_\mu, \mathcal{H})$ is $\mathcal{N}\mu\mathcal{H}$-CC and $\{U_\lambda : \lambda \in \Lambda\}$ is the $\mu_\theta$-open cover of $X$. Then, for all $x \in X$, there exists $\lambda_x \in \Lambda$ where $x \in U_{\lambda_x}$. Since $U_{\lambda_x}$ is $\mu_\theta$-open, then there exists $M_x \in \mu$ where $x \in M_x \subset Cl_\mu(M_x) \subset U_{\lambda_x}$. However, $M_x \subseteq Int_\mu Cl_\mu(M_x) \subseteq Cl_\mu(M_x)$. Then, $X = \bigcup_{X_n \in X} M_{x_n}$ where $n \in \mathbb{N}$. Since $X$ is $\mathcal{N}\mu\mathcal{H}$-CC, there exist $x_1, x_2, ..., x_n \in X$ where $X \backslash \bigcup_{k=1}^{n} Int_\mu(Cl_\mu(M_{x_k})) \in \mathcal{H}$. However, $X \backslash \bigcup_{k=1}^{n} (U_{\lambda_{x_k}}) \subset X \backslash \bigcup_{k=1}^{n} Int_\mu(Cl_\mu(M_{x_k})) \in \mathcal{H}$. Hence, $X \backslash \bigcup_{k=1}^{n} (U_{\lambda_{x_k}}) \in \mathcal{H}$. $\square$

**Theorem 4.** *Let $X_\mu$ be a $\mu$-regular space. The following statements are equivalent:*

1. *$(X_\mu, \mathcal{H})$ is $\mathcal{N}\mu\mathcal{H}$-CC.*
2. *$(X_\mu, \mathcal{H})$ is $\mu\mathcal{H}$-CC.*

**Proof.** $(1) \Rightarrow (2) : \cdot$ Suppose $X$ is $\mu$-regular and $\mathcal{N}\mu\mathcal{H}$-CC and $\{U_\lambda : \lambda \in \Lambda\}$ is the $\mu_\theta$-open cover of $X$. Then, for all $x \in X$, there exists $\lambda_x \in \Lambda$ where $x \in U_{\lambda_x}$. Since $U_{\lambda_x}$ is $\mu_\theta$-open, then there exists $M_x \in \mu$ such that $x \in M_x \subset Cl_\mu(M_x) \subset U_{\lambda_x}$. However, $M_x \subseteq Int_\mu(Cl_\mu(M_x)) \subseteq Cl_\mu(M_x)$. Then, the sub-collection $\{M_{x_n} : x \in X\}$ is the $\mu$-open cover of

$X$. Since $X$ is $\mathcal{N}\mu\mathcal{H}$-CC, so there exist $x_1, x_2, ..., x_n \in X$ where $X \backslash \bigcup\limits_{k=1}^{n} Int_\mu(Cl_\mu(M_{x_k})) \in \mathcal{H}$. However, $X \backslash \bigcup\limits_{k=1}^{n} (U_{\lambda_{x_k}}) \subset X \backslash \bigcup\limits_{k=1}^{n} Int_\mu(Cl_\mu(M_{x_k})) \in \mathcal{H}$. Thus, $X \backslash \bigcup\limits_{k=1}^{n} (U_{\lambda_{x_k}}) \in \mathcal{H}$. This mean $(X_\mu, \mathcal{H})$ is $\mu\mathcal{H}$-CC.

$(2) \Rightarrow (1) : \cdot$ It follows from Theorem 2. $\square$

## 3. Weakly $\mu$-Countably Compactness and Weakly $\mu\mathcal{H}$-Countably Compactness

In this section, we introduce the notion of weakly $\mu$-countably compactness and the notion of weakly $\mu\mathcal{H}$-countably compactness. We also present a diagram to describe the relationships among different types of generalizations of $\mu$-compactness and $\mu\mathcal{H}$-compactness.

**Definition 14.** *Let $X$ be a set. The space $X_\mu$ is said to be weakly $\mu$-countably compact (denoted by $\mathcal{W}\mu$- CC) whenever $X = \bigcup\limits_{\lambda \in \Lambda} U_\lambda$, where $U_\lambda \in \mu$ for all $\lambda \in \Lambda$ and $\Lambda$ is a countable index set, then there is a finite sub-collection $\{U_\lambda : \lambda \in \Lambda_0 \subseteq \Lambda\}$ such that $X = \bigcup\limits_{\lambda \in \Lambda_0} Cl_\mu(U_\lambda)$.*

**Theorem 5.** *A space $X_\mu$ is $\mathcal{W}\mu$-CC if and only if whenever $X = \bigcup\limits_{\lambda \in \Lambda} U_\lambda$, where $U_\lambda$ is a $\mu$-regular open subset for all $\lambda \in \Lambda$, then there exists a finite subset $\Lambda_0 \subset \Lambda$ such that $X = \bigcup\limits_{\lambda \in \Lambda_0} Cl_\mu(U_\lambda)$.*

**Proof.** Necessity. It is straightforward and therefore omitted.

Sufficiency. Suppose $X = \bigcup\limits_{\lambda \in \Lambda} U_\lambda$, where $U_\lambda \in \mu$ for all $\lambda \in \Lambda$ and $\Lambda$ is a countable index set. It is clear that $Int_\mu Cl_\mu(U_\lambda)$ is $\mu$-open, thus $\mathcal{Z} = \{Int_\mu Cl_\mu(U_\lambda) : \lambda \in \Lambda\}$ is a countable $\mu$-regular open cover of $X$. So we can find a finite sub-collection $\{U_\lambda : \lambda \in \Lambda_0 \subseteq \Lambda\}$ of $X$ where $X = \bigcup\limits_{\lambda \in \Lambda_0} Cl_\mu(Int_\mu Cl_\mu(U_\lambda))$. It is clear that $Cl_\mu(Int_\mu Cl_\mu(U_\lambda))$ is $\mu$-closed, thus $X = \bigcup\limits_{\lambda \in \Lambda_0} Cl_\mu(U_\lambda)$. Hence, $X_\mu$ is $\mathcal{W}\mu$-CC. $\square$

**Theorem 6.** *Let $X_\mu$ be a space. The following statements are equivalent:*

1. *$X$ is $\mathcal{W}\mu$-CC;*
2. *For any countable collection $\mathcal{F} = \{U_\lambda : \lambda \in \Lambda\}$ of countable $\mu$-closed subset of $X$ such that $\bigcap\limits_{\lambda \in \Lambda_0} U_\lambda = \varnothing$, there exists a finite sub-collection $\{U_\lambda : \lambda \in \Lambda_0 \subseteq \Lambda\}$ such that $\bigcap\limits_{\lambda \in \Lambda_0} Int_\mu(U_\lambda) = \varnothing$;*
3. *For any countable collection $\mathcal{F} = \{U_\lambda : \lambda \in \Lambda\}$ of countable $\mu$-regular closed subsets of $X$ such that $\bigcap\limits_{\lambda \in \Lambda_0} U_\lambda = \varnothing$, there exists a finite sub-collection $\{U_\lambda : \lambda \in \Lambda_0 \subseteq \Lambda\}$ such that $\bigcap\limits_{\lambda \in \Lambda_0} Int_\mu(U_\lambda) = \varnothing$.*

**Proof.** $(1) \Rightarrow (2) : \cdot$ Suppose $X$ is $\mathcal{W}\mu$-CC and $\mathcal{F} = \{U_\lambda : \lambda \in \lambda\}$ is a countable sub-collection of a $\mu$-closed subset of $X$ such that $\bigcap\{U_\lambda : \lambda \in \Lambda\} = \varnothing$. Then, $X = X \backslash \bigcap \mathcal{F} = \bigcup X \backslash \mathcal{F}$. Since $X$ is $\mathcal{W}\mu$-CC, there exists a finite sub-collection $\{X \backslash U_\lambda : \lambda \in \Lambda_0 \subseteq \Lambda\}$ cover of $X$. Thus, $X = \bigcup\limits_{\lambda \in \Lambda_0} Cl_\mu(X \backslash U_\lambda)$. Hence,

$X \backslash \bigcup\limits_{\lambda \in \Lambda_0} Cl_\mu(X \backslash U_\lambda) = X \backslash Cl_\mu(\bigcup\limits_{\lambda \in \Lambda_0} (X \backslash U_\lambda)) = Int_\mu(X \backslash (\bigcup\limits_{\lambda \in \Lambda_0} (X \backslash U_\lambda)))$

$= \bigcap\limits_{\lambda \in \Lambda_0} Int_\mu(U_\lambda)$. Thus, $\bigcap\limits_{\lambda \in \Lambda_0} Int_\mu(U_\lambda) = \varnothing$

$(2) \Rightarrow (1) : \cdot$ Suppose $\{U_\lambda : \lambda \in \Lambda\}$ is a countable of $\mu$-open cover of $X$. Thus, $\{X \backslash U_\lambda : \lambda \in \Lambda\}$ is a countable of $\mu$-closed subset of $X$.

Since $X = \bigcup\limits_{\lambda \in \Lambda} (U_\lambda)$, so $X \backslash \bigcup\limits_{\lambda \in \Lambda} (U_\lambda) = \bigcap\limits_{\lambda \in \Lambda} (X \backslash U_\lambda) = \varnothing$. So, by the assumption that there exists a finite sub-collection $\{X \backslash U_\lambda : \lambda \in \Lambda_0\}$ of $\mathcal{F}$ such that

$Int_\mu(( \bigcap_{\lambda \in \Lambda_0} (X \backslash U_\lambda)) = \emptyset$.

Hence, $X = X \backslash Int_\mu((\bigcap_{\lambda \in \Lambda_0} (X \backslash U_\lambda)) = Cl_\mu(X \backslash \bigcap_{\lambda \in \Lambda_0} (X \backslash U_\lambda) = (\bigcup_{\lambda \in \Lambda_0} Cl_\mu(U_\lambda))$. Therefore, $X$ is $\mathcal{W}\mu$-CC.

$(3) \Rightarrow (1) : \cdot$ Suppose $\{U_\lambda : \lambda \in \Lambda\}$ is a countable $\mu$-open cover of $X$ and so $\{Int_\mu(Cl_\mu(U_\lambda)) : \lambda \in \Lambda\}$ is a countable $\mu$-regular open cover of $X$.

Thus, $\{X \backslash Int_\mu(Cl_\mu(U_\lambda)) : \lambda \in \Lambda\}$ is a $\mu$-regular closed subset of $X$ such that $X \backslash \bigcup_{\lambda \in \Lambda} Int_\mu(Cl_\mu(U_\lambda)) = \bigcap_{\lambda \in \Lambda} Cl_\mu(Int_\mu(X \backslash U_\lambda) = \emptyset$, so by the assumption that there exists a finite sub-collection $\{U_\lambda : \lambda \in \Lambda_0 \subseteq \Lambda\}$ of $\mathcal{F}$ such that $Int_\mu((\bigcap_{\lambda \in \Lambda_0} Cl_\mu(Int_\mu(X \backslash U_\lambda)) = \emptyset$.

Hence, $X = X \backslash Int_\mu(\bigcap_{\lambda \in \Lambda_0} (Cl_\mu(Int_\mu(X \backslash U_\lambda))) = Cl_\mu(X \backslash \bigcap_{\lambda \in \Lambda_0} (X \backslash U_\lambda) = (\bigcup_{\lambda \in \Lambda_0} Cl_\mu(U_\lambda))$. It is clear that $X$ is $\mathcal{W}\mu$-CC.

$(2) \Leftrightarrow (3) : \cdot$ It is obvious since $\mu$-regular closed is $\mu$-closed.

$(1) \Rightarrow (3) : \cdot$ It is similar to $(1) \Rightarrow (2) :$ since $\mu$-regular closed is $\mu$-closed. $\square$

**Theorem 7.** *If a space $X_\mu$ is $\mathcal{W}\mu$-CC, then every countable cover of $X$ by $\mu_\theta$-open sets has a finite sub-cover.*

**Proof.** Suppose $X_\mu$ is $\mathcal{W}\mu$-CC and $\mathcal{F} = \{U_\lambda : \lambda \in \Lambda\}$ be $\mu_\theta$-open countable cover of $X$. Then, for all $x \in X$, there exists $\lambda_x \in \Lambda$ such that $x \in U_{\lambda_x}$. Since $U_{\lambda_x}$ is a $\mu_\theta$-open, then there exists $M_x \in \mu$ where $x \in M_x \subset Cl_\mu(M_x) \subset U_{\lambda_x}$. However, $X$ is $\mathcal{W}\mu$-CC, so there exist $x_1, x_2, ..., x_n \in X$ where $X = \bigcup_{k=1}^{n} Cl_\mu(M_{x_k}) = \bigcup_{k=1}^{n} (U_{\lambda_{x_k}})$. $\square$

**Theorem 8.** *Let $X_\mu$ be a $\mu$-regular space. Then, $X_\mu$ is $\mathcal{W}\mu$-CC if and only if $X_\mu$ is $\mu$-CC.*

**Proof.** It is straightforward and therefore omitted. $\square$

**Definition 15.** *Let $X_\mu$ be a space. A subset $A$ of $X$ is said to be weakly $\mu$-countably compact set (denoted by $\mathcal{W}\mu$-CC set) whenever $A \subset \bigcup_{\lambda \in \Lambda} U_\lambda$, where $U_\lambda \in \mu$ for all $\lambda \in \Lambda$ and $\Lambda$ is a countable index set, then there is a finite sub-collection $\{U_\lambda : \lambda \in \Lambda_0 \subseteq \Lambda\}$ such that $A \subset \bigcup_{\lambda \in \Lambda_0} Cl_\mu(U_\lambda)$.*

**Theorem 9.** *A subset $A$ of $X_\mu$ is $\mathcal{W}\mu$-CC set if and only if, whenever $A = \bigcup_{\lambda \in \Lambda} U_\lambda$, where $U_\lambda$ is $\mu$-regular open subset for all $\lambda \in \Lambda$, then there exists a finite sub-collection $\{U_\lambda : \lambda \in \Lambda_0 \subseteq \Lambda\}$ such that $A = \bigcup_{\lambda \in \Lambda} Cl_\mu(U_\lambda)$.*

**Proof.** It is straightforward and therefore omitted. $\square$

**Theorem 10.** *Let $A$ be a subset of $X_\mu$. The following statements are equivalent:*

1. *$A$ is $\mathcal{W}\mu$-CC;*
2. *For any countable collection $\mathcal{F} = \{U_\lambda : \lambda \in \Lambda\}$ of a $\mu$-closed subset of $X$ such that $[\bigcap\{U_\lambda : \lambda \in \Lambda\}] \cap A = \emptyset$, there exists a finite sub-collection $\Lambda_0 \in \Lambda$ of $\mathcal{F}$ such that $[\bigcap_{\lambda \in \Lambda_0} Int_\mu(U_\lambda)] \cap A = \emptyset$;*
3. *For any countable collection $\mathcal{F} = \{U_\lambda : \lambda \in \Lambda\}$ of $\mu$-regular closed subsets of $X$ such that $[\bigcap\{U_\lambda : \lambda \in \Lambda\}] \cap A = \emptyset$, there exists a finite sub-collection $\Lambda_0 \in \Lambda$ of $\mathcal{F}$ such that $[\bigcap_{\lambda \in \Lambda_0} Int_\mu(U_\lambda)] \cap A = \emptyset$.*

**Proof.** $(1) \Rightarrow (2) : \cdot$ Suppose $A$ is $\mathcal{W}\mu$-CC set and $\mathcal{F} = \{U_\lambda : \lambda \in \Lambda\}$ is a $\mu$-closed countable collection of $X$ such that $\bigcap\{U_\lambda : \lambda \in \Lambda\} \cap A = \emptyset$. Then, $A \subseteq X \backslash \bigcap \mathcal{F} = \bigcup X \backslash \mathcal{F}$. Since $X$ is $\mathcal{W}\mu$-CC, there exists a finite sub-collection $\{U_\lambda : \lambda \in \Lambda_0 \subseteq \Lambda\}$ cover of $A$ such

that $\{X\backslash U_\lambda : \lambda \in \Lambda_0 \in \Lambda\}$. Thus, $A \subseteq \bigcup\limits_{\lambda \in \Lambda_0} Cl_\mu(X\backslash U_\lambda)$. Hence, $X\backslash \bigcup\limits_{\lambda \in \Lambda_0} Cl_\mu(X\backslash U_\lambda) =$

$X\backslash Cl_\mu(\bigcup\limits_{\lambda \in \Lambda_0}(X\backslash U_\lambda)) = Int_\mu(X\backslash(\bigcup\limits_{\lambda \in \Lambda_0}(X\backslash U_\lambda))$

$= \bigcap\limits_{\lambda \in \Lambda_0} Int_\mu(U_\lambda)$. Thus, $[\bigcap\limits_{\lambda \in \Lambda_0} Int_\mu(U_\lambda)] \cap A = \varnothing$

$(2) \Rightarrow (1) : \cdot$ Suppose $\{U_\lambda : \lambda \in \Lambda\}$ is a countable $\mu$-open cover of $A$. Thus, $\{X\backslash U_\lambda : \lambda \in \Lambda\}$ is a $\mu$-closed subset of $X$. By the assumption that $X\backslash \bigcup\limits_{\lambda \in \Lambda}(U_\lambda) \cap A = \bigcap\limits_{\lambda \in \Lambda}(X\backslash U_\lambda) \cap A = \varnothing$, so there exists a finite sub-collection $\Lambda_0 \in \Lambda$ of $\mathcal{F}$ such that

$Int_\mu((\bigcap\limits_{\lambda \in \Lambda_0}(X\backslash U_\lambda)) = \varnothing$.

Hence, $A \subseteq X\backslash Int_\mu((\bigcap\limits_{\lambda \in \Lambda_0}(X\backslash U_\lambda)) = Cl_\mu(X\backslash \bigcap\limits_{\lambda \in \Lambda_0}(X\backslash U_\lambda) = (\bigcup\limits_{\lambda \in \Lambda_0} Cl_\mu(U_\lambda))$. Therefore,

$X$ is $\mathcal{W}\mu$-CC.

$(3) \Rightarrow (1) : \cdot$ Suppose $A = \bigcup\limits_{\lambda \in \Lambda} U_\lambda$ where $U_\lambda \in \mu$ for all $\lambda \in \Lambda$ and $\Lambda$ is a countable index

set, so $A = \bigcup\limits_{\lambda \in \Lambda} Int_\mu(Cl_\mu(U_\lambda))$. Thus, $\{X\backslash Int_\mu(Cl_\mu(U_\lambda)) : \lambda \in \Lambda\}$ is a $\mu$-regular closed

subset of $X$. By the assumption that $X\backslash \bigcup\limits_{\lambda \in \Lambda} Int_\mu(Cl_\mu(U_\lambda)) \cap A = \bigcap\limits_{\lambda \in \Lambda} Cl_\mu(Int_\mu(X\backslash U_\lambda) \cap$

$A = \varnothing$, so there exists a finite sub-collection $\{U_\lambda : \lambda \in \Lambda_0 \subseteq \Lambda\}$ of $\mathcal{F}$ such that

$Int_\mu((\bigcap\limits_{\lambda \in \Lambda_0} Cl_\mu(Int_\mu(X\backslash U_\lambda)) = \bigcap\limits_{\lambda \in \Lambda_0} Int_\mu(Cl_\mu(Int_\mu(X\backslash U_\lambda))) = \varnothing$.

Hence,

$A \subseteq X\backslash \bigcap\limits_{\lambda \in \Lambda_0} Int_\mu(Cl_\mu(Int_\mu(X\backslash U_\lambda))) = Cl_\mu(X\backslash \bigcap\limits_{\lambda \in \Lambda_0}(X\backslash U_\lambda) = (\bigcup\limits_{\lambda \in \Lambda_0} Cl_\mu(U_\lambda))$. It is clear

that $A$ is $\mathcal{W}\mu$-CC set.

$(2) \Leftrightarrow (3) : \cdot$ It is obvious since $\mu$-regular closed is $\mu$-closed.

$(1) \Rightarrow (3) : \cdot$ It is similar to $(1) \Rightarrow (2) :$ since $\mu$-regular closed is $\mu$-closed. $\quad\square$

**Theorem 11.** *Let $A$ be a $\mathcal{W}\mu$-CC subset of a space $X_\mu$. Then, every cover of $A$ by $\mu_\theta$-open subsets of $X$ has a finite subcover.*

**Proof.** It is straightforward and therefore omitted. $\quad\square$

**Theorem 12.** *Let $A, B \subseteq X_\mu$ and $X\backslash A$ be countable. If $A$ is $\mu_\theta$-closed and $B$ is $\mathcal{W}\mu$-CC, then $A \cap B$ is $\mathcal{W}\mu$-CC set.*

**Proof.** Let $A \cap B \subseteq \bigcup\limits_{\lambda \in \Lambda} U_\lambda$, where $U_\lambda \in \mu$ for all $\lambda \in \Lambda$ is a countable index set,

and $\mathcal{F} = \{U_\lambda : \lambda \in \Lambda\}$. Then, $B \subseteq (\bigcup\limits_{\lambda \in \Lambda} U_\lambda)\bigcup(X\backslash A)$. Additionally, for all $x \notin A$,

there exists $U_x \in \mu$ where $x \in U_x \subset Cl_\mu(U_x) \subset X\backslash A$. Since $U_x$ is a $\mu_\theta$-open and $X\backslash A$ is countable, then $\mathcal{F} \cup \{U_x : x \in X\backslash A\}$ is a countable $\mu$-open cover of $B$. However, $B$ is $\mathcal{W}\mu$-CC, so there exist $\lambda_1, \lambda_2, ..., \lambda_n \in \Lambda$ and there exist $x_1, x_2, ..., x_m \in X\backslash A$ such that $B \subseteq (\bigcup\limits_{k=1}^{n} Cl_\mu(U_{\lambda_k}))\bigcup(\bigcup\limits_{k=1}^{m} Cl_\mu(U_{x_k}))$. However, $Cl_\mu(U_{x_k}) \subset X\backslash A$, thus $A \cap B \subseteq \bigcup\limits_{k=1}^{n} Cl_\mu(U_{\lambda_k})$.

Hence, $A \cap B$ is a $\mathcal{W}\mu$-CC set. $\quad\square$

**Theorem 13.** *Let $A \subseteq B \subseteq X_\mu$ . If $A$ is $\mathcal{W}\mu_B$-CC, then $A$ is $\mathcal{W}\mu$-CC set.*

**Proof.** Suppose that $A$ is $\mathcal{W}\mu_B$-CC set, and $\mathcal{U} = \{U_\lambda : \lambda \in \Lambda\}$ is a countable $\mu$-open cover of $A$. Then, $\mathcal{U}_\mathcal{B} = \{U_\lambda : \lambda \in \Lambda\}$ is a $\mu_B$-open cover of $A$. However, $A$ is $\mathcal{W}\mu_B$-CC, so there exists a finite sub-collection $\{U_\lambda : \lambda \in \Lambda_0 \subseteq \Lambda\}$ of $\mathcal{U}_\mathcal{B}$ such that $A = \bigcup\limits_{\lambda \in \Lambda_0} Cl_{\mu_B}(U_\lambda \cap B)$.

It is clear that $Cl_{\mu_B}(U_\lambda \cap B) = (Cl_\mu(U_\lambda \cap B)) \cap B \subset Cl_\mu(U_\lambda)$ where $\lambda \in \Lambda_0$. Hence, $A$ is $\mathcal{W}\mu$-CC set. $\quad\square$

**Definition 16.** *Let* $(X_\mu, \mathcal{H})$ *be a space with respect to* $\mathcal{H}$. *The pair* $(X_\mu, \mathcal{H})$ *is said to be weakly* $\mu\mathcal{H}$-*countably compact (denoted by* $\mathcal{W}\mu\mathcal{H}$- *CC) whenever* $X = \bigcup\limits_{\lambda \in \Lambda} U_\lambda$, *where* $U_\lambda \in \mu$ *for all* $\lambda \in \Lambda$ *and* $\Lambda$ *is a countable index set, then there is a finite sub-collection* $\{U_\lambda : \lambda \in \Lambda_0 \subseteq \Lambda\}$ *such that* $X \backslash \bigcup\limits_{\lambda \in \Lambda_0} Cl_\mu(U_\lambda) \in \mathcal{H}$.

**Example 4.** *Let* $X = (0,1)$, $\mu = \{\phi, G_n : n \in \mathbb{Z}^+\}$, *where* $G_n = (\frac{1}{n}, 1)$ *and* $\mathcal{H} = \mathcal{H}_f$. *Then,* $X_\mu$ *is* $\mathcal{N}\mu\mathcal{H}$-*CC because for any proper* $\mu$-*open set* $Int_\mu Cl_\mu(G_{n_i}) = X$ *where* $i \in \mathbb{Z}^+$, *then* $X \backslash \bigcup\limits_{i}^{n} Int_\mu Cl_\mu(G_{n_i}) \in \mathcal{H}$. *However, that is not* $\mu\mathcal{H}$-*CC because there is no finite sub-collection such that* $X \backslash \bigcup\limits_{k}^{n} G_{n_i} \in \mathcal{H}$.

**Example 5.** *Let* $X = \mathbb{Z}$, $\mathcal{K} = \{\{2n-1, 2n, 2n+1\} : n \in \mathbb{Z}\}$, *and* $\mu$ *generated by* $\mu$-*subbase* $\mathcal{S}$ *and* $\mathcal{H} = \mathcal{P}(\mathbb{Z})$. *Then,* $(X_{\mu(\mathcal{K})}, \mathcal{H})$ *is* $\mathcal{W}\mu\mathcal{H}$-*CC, but not* $\mathcal{W}\mu$-*CC.*

**Theorem 14.** *A space* $(X_\mu, \mathcal{H})$ *with respect to* $\mathcal{H}$ *is* $\mathcal{W}\mu\mathcal{H}$-*CC if and only if for any countable* $\mu$-*regular open cover* $\{U_\lambda : \lambda \in \Lambda\}$ *of* $X$, *there exits a finite sub-collection* $\{U_\lambda : \lambda \in \Lambda_0 \subseteq \Lambda\}$ *such that* $X \backslash \bigcup\limits_{\lambda \in \Lambda_0} Cl_\mu(U_\lambda) \in \mathcal{H}$.

**Proof.** Necessity. It is straightforward and therefore omitted.
Sufficiency. Let $X = \bigcup\limits_{\lambda \in \Lambda} U_\lambda$, where $U_\lambda \in \mu$ for all $\lambda \in \Lambda$ and $\Lambda$ is a countable index set. It is clear that $Int_\mu(Cl_\mu(U_\lambda))$ is $\mu$-open, thus $\mathcal{Z} = \{Int_\mu(Cl_\mu(U_\lambda)) : \lambda \in \Lambda\}$ bis a countable $\mu$-regular open cover of $X$. Then, there exists a finite sub-collection $\{U_\lambda : \lambda \in \Lambda_0 \subseteq \Lambda\}$ such that $X \backslash \bigcup\limits_{\lambda \in \Lambda_0} Cl_\mu(Int_\mu(Cl_\mu(U_\lambda))) \in \mathcal{H}$.
However, $X \backslash \bigcup\limits_{\lambda \in \Lambda_0} Cl_\mu(U_\lambda) \subseteq X \backslash \bigcup\limits_{\lambda \in \Lambda_0} Cl_\mu(Int_\mu Cl_\mu(U_\lambda))$. Thus, $X \backslash \bigcup\limits_{\lambda \in \Lambda_0} Cl_\mu(U_\lambda) \in \mathcal{H}$.
Hence, $X_\mu$ is $\mathcal{W}\mu$-CC. $\square$

**Theorem 15.** *If a space* $(X_\mu, \mathcal{H})$ *is* $\mathcal{W}\mu\mathcal{H}$-*CC, then for every countable cover of* $X$ *by* $\mu_\theta$-*open sets there exists a finite sub-collection* $\{U_\lambda : \lambda \in \Lambda_0 \subseteq \Lambda\}$ *such that* $X \backslash \bigcup\limits_{\lambda \in \Lambda_0} (U_\lambda) \in \mathcal{H}$.

**Proof.** Suppose $(X_\mu, \mathcal{H})$ is $\mathcal{W}\mu\mathcal{H}$-CC and $\{U_\lambda : \lambda \in \Lambda\}$ be a $\mu_\theta$-open cover of $X$. Then, for all $x \in X$, there exists $\lambda_x \in \Lambda$ such that $x \in U_{\lambda_x}$. Thus, there exists $M_x \in \mu$ such that $x \in M_x \subset Cl_\mu(M_x) \subset U_{\lambda_x}$. Then, $X = \bigcup\limits_{x \in X} M_{x_n}$ where $n \in \mathbb{N}$. Since $X$ is $\mathcal{W}\mu\mathcal{H}$-CC, so there exist $x_1, x_2, ..., x_n \in X$ where $X \backslash \bigcup\limits_{k=1}^{n} Cl_\mu(M_{x_k}) \in \mathcal{H}$. However, $X \backslash \bigcup\limits_{k=1}^{n} (U_{\lambda_{x_k}}) \subseteq X \backslash \bigcup\limits_{k=1}^{n} Cl_\mu(M_{x_k}) \in \mathcal{H}$. Hence, $X \backslash \bigcup\limits_{k=1}^{n} (U_{\lambda_{x_k}}) \in \mathcal{H}$. $\square$

**Theorem 16.** *Let* $X_\mu$ *be a* $\mu$-*regular space. The following statements are equivalent:*

1. $(X_\mu, \mathcal{H})$ *is* $\mathcal{W}\mu\mathcal{H}$-*CC;*
2. $(X_\mu, \mathcal{H})$ *is* $\mu\mathcal{H}$-*CC.*

**Proof.** $(1) \Rightarrow (2) :$ · Suppose $X$ is a $\mu$-regular, and $\mathcal{W}\mu\mathcal{H}$-CC and $\{U_\lambda : \lambda \in \Lambda\}$ are $\mu_\theta$-open covers of $X$. Then, for all $x \in X$, there exists $\lambda_x \in \Lambda$ such that $x \in U_{\lambda_x}$. Thus, there exists $M_x \in \mu$ where $x \in M_x \subset Cl_\mu(M_x) \subset U_{\lambda_x}$. Then, the sub-collection $\{M_{x_n} : x \in X\}$ is a countable $\mu$-open cover of $X$. Since $X$ is $\mathcal{W}\mu\mathcal{H}$-CC, so there exist $x_1, x_2, ..., x_n \in X$ where $X \backslash \bigcup\limits_{k=1}^{n} Cl_\mu(M_{x_k}) \in \mathcal{H}$. However, $X \backslash \bigcup\limits_{k=1}^{n} (U_{\lambda_{x_k}}) \subseteq X \backslash \bigcup\limits_{k=1}^{n} Cl_\mu(M_{x_k}) \in \mathcal{H}$. Thus, $X \backslash \bigcup\limits_{k=1}^{n} (U_{\lambda_{x_k}}) \in \mathcal{H}$. This means $(X_\mu, \mathcal{H})$ is $\mu\mathcal{H}$-CC.

$(2) \Rightarrow (1) : \cdot$ It is clear that $X \backslash \bigcup\limits_{k=1}^{n} Cl_\mu(M_{x_k}) \subseteq X \backslash \bigcup\limits_{k=1}^{n} (M_{x_k}) \in \mathcal{H}.$

Thus, $X \backslash \bigcup\limits_{k=1}^{n} (Cl_\mu(M_{x_k}) \in \mathcal{H}.$   $\square$

**Theorem 17.** *Let $A$ be a $\mathcal{W}\mu\mathcal{H}$-CC, then for every countable cover of $A$ by $\mu_\theta$-open sets there exits a finite sub-collection $\{U_\lambda : \lambda \in \Lambda_0 \subseteq \Lambda\}$ such that $A \backslash \bigcup\limits_{\lambda \in \Lambda_0} Cl_\mu(U_\lambda) \in \mathcal{H}.$*

**Theorem 18.** *Let $A, B \subseteq X_\mu$ be subsets of a space $X_\mu$ and $X \backslash A$ is countable. If $A$ is $\mu_\theta$-closed and $B$ is $\mathcal{W}\mu\mathcal{H}$-CC, then $A \cap B$ is $\mathcal{W}\mu\mathcal{H}$-CC.*

**Proof.** Let $\mathcal{F} = \{U_\lambda : \lambda \in \Lambda\}$ be a countable $\mu$-open cover of $A \cap B$. Then, $\mathcal{F} \cup X \backslash A$ is a countable $\mu$-open cover $B$. Since $X \backslash A$ is a $\mu_\theta$-open for all $x \notin A$, there exists a $\mu$-open set $U_x$ where $x \in U_x \subset Cl_\mu(U_x) \subset X \backslash A$. Thus, $\mathcal{F} \cup \{U_x : x \in X \backslash A\}$ is a countable $\mu$-open cover of $B$. However, $B$ is $\mathcal{W}\mu$-CC, so there exist $\lambda_1, \lambda_2, ..., \lambda_n \in \Lambda$ and $x_1, x_2, ..., x_m \in X \backslash A$ where $B \backslash (\bigcup\limits_{k=1}^{n} Cl_\mu(U_{\lambda_k})) \bigcup (\bigcup\limits_{k=1}^{m} Cl_\mu(U_{x_k})) \in \mathcal{H}.$

Thus, $A \cap B \backslash (\bigcup\limits_{k=1}^{n} Cl_\mu(U_{\lambda_k})) \bigcup (\bigcup\limits_{k=1}^{m} Cl_\mu(U_{x_k})) \subset B \backslash (\bigcup\limits_{k=1}^{n} Cl_\mu(U_{\lambda_k})) \bigcup (\bigcup\limits_{k=1}^{m} Cl_\mu(U_{x_k}))$. Hence, $A \cap B \backslash (\bigcup\limits_{k=1}^{n} Cl_\mu(U_{\lambda_k})) \bigcup (\bigcup\limits_{k=1}^{m} Cl_\mu(U_{x_k})) \in \mathcal{H}.$ This mean $A \cap B$ is $\mathcal{W}\mu\mathcal{H}$-CC.   $\square$

**Theorem 19.** *Let $(X_\mu, \mathcal{H})$ be a space with respect to $\mathcal{H}$ where $\mathcal{H}$ is an ideal on $X$, then the union of two $\mathcal{W}\mu\mathcal{H}$-CC sets is a $\mathcal{W}\mu\mathcal{H}$-CC set.*

**Proof.** Suppose $A$ and $B$ are $\mathcal{W}\mu\mathcal{H}$-CC sets of $X$. Let $\mathcal{F} = \{U_\lambda : \lambda \in \Lambda\}$ be any countable $\mu$-open cover of $A \cup B$ of $X$, then there exist finite subsets $\Lambda_0, \Lambda_1 \subseteq \Lambda$ where $A \backslash \bigcup\limits_{\Lambda_0 \in \Lambda} (U_\lambda) \in \mathcal{H}$ and $B \backslash \bigcup\limits_{\Lambda_1 \in \Lambda} (U_\lambda) \in \mathcal{H}.$

Thus, $A \cup B \backslash \bigcup\limits_{\lambda \in \Lambda_0 \cup \Lambda_1} (U_\lambda) \subset (A \backslash \bigcup\limits_{\Lambda_0 \in \Lambda} (U_\lambda)) \bigcup (B \backslash \bigcup\limits_{\Lambda_1 \in \Lambda} (U_\lambda))$. However, $\Lambda_0 \cup \Lambda_1$ is a finite subset of $\Lambda$ and $\mathcal{H}$ is an ideal on $X$. Then, $A \cup B \backslash \bigcup\limits_{\lambda \in \Lambda_0 \cup \Lambda_1} (U_\lambda) \in \mathcal{H}.$ Hence, $A \cup B$ is $\mathcal{W}\mu\mathcal{H}$-CC.   $\square$

Example 6 illustrates that $\mathcal{H}$ being an ideal is a necessary condition.

**Example 6.** *Let $X = \mathbb{N}$, $\mu = \mathcal{P}(\mathbb{N})$, and hereditary class $\mathcal{H} = \{A \subset \mathbb{N} : A$ is subset of the set of all odd numbers or $A$ is a subset of the set of all even numbers $\}$. Let $A$ be the set of all odd numbers and $B$ be the set of all even numbers, then $A$ and $B$ are $\mathcal{W}\mu\mathcal{H}$-CC sets. While $A \cup B$ is not $\mathcal{W}\mu\mathcal{H}$-CC. Let $\bigcup\limits_{n \in \mathbb{N}} \{2n-1, 2n\}\} = A \cup B$ where $\{2n-1, 2n\} \in \mu$ for all $n \in \mathbb{N}$. Thus, $(A \cup B) \backslash \bigcup\limits_{k=1}^{m} Cl_\mu(\{2n_k - 1, 2n_k\}) \notin \mathcal{H}$, for some $n_k$, where $k = 1, 2, ..., m.$*

**Theorem 20.** *Let $X_\mu$ be a $\mathcal{N}\mu$-CC space, then $X_\mu$ is a $\mathcal{W}\mu$-CC space.*

**Proof.** Suppose $X_\mu$ is a $\mathcal{N}\mu$-CC space. Then, for each countable $\mu$-open cover $\{U_\lambda : \lambda \in \Lambda\}$ of $X$, there exists a finite sub-collection $\{U_\lambda : \lambda \in \Lambda_0 \subseteq \Lambda\}$ of $X$ such that $X = \bigcup\limits_{\lambda \in \Lambda_0} Int_\mu Cl_\mu(U_\lambda)$. However, $Int_\mu Cl_\mu(U_\lambda) \subseteq Cl_\mu(U_\lambda)$.

Thus, $X = \bigcup\limits_{\lambda \in \Lambda_0} Int_\mu Cl_\mu(U_\lambda) \subseteq \bigcup\limits_{\lambda \in \Lambda_0} Cl_\mu(U_\lambda)$. Hence, $X = \bigcup\limits_{\lambda \in \Lambda_0} Cl_\mu(U_\lambda).$   $\square$

**Lemma 3.** *Let $X_\mu$ be a space such that $X = [0, 1] \subseteq \mathbb{R}$, and $X_1, X_2, X_3$ be disjoint dense $\mu$-subspaces of $X$ such that $X = X_1 \cup X_2 \cup X_3$. Consider the $\mu^* = \{\emptyset, X, X_1, X_2, X_1 \cup X_2\}$ and*

$\Psi = \mu \wedge \mu^*$ *generated by the finite intersection of elements of $\mu$ and $\mu^*$, then if $C$ is a $\mu$-regular closed subset of $X_\Psi$ and $A$ is a $\mu$-open subset of $X_\mu$ such that $C \subseteq A$, then $Int_\Psi(C) \subseteq Int_\mu Cl_\Psi(A)$*

**Proof.** It is straightforward and therefore omitted. □

The converse of Theorem 20 is not true, as illustrated in Example 7.

**Example 7.** *Let $X_\mu$ and $X_\Psi$ as they are in the above Lemma 3.20. It is proved that $X_\Psi$ is not almost compact in [23], so it is not nearly $\mu$-CC. We prove that $X_\Psi$ is weakly $\mu$-CC. Let $\{U_\lambda : \lambda \in \Lambda\}$ be a countable $\mu$-regular open cover of $X_\Psi$, so there is $C_\lambda$ $\mu$-regular closed in $X_\Psi$ where $Int_\Psi(C_\lambda) \subseteq C_\lambda \subseteq U_\lambda$ and $X = \bigcup_{\lambda \in \Lambda} Int_\Psi(C_\lambda))$. Then, by Lemma 3.20, we obtain $Int_\Psi(C_\lambda) \subseteq Int_\mu(Cl_\Psi(U_\lambda))$, then $X_\mu = \bigcup_{\lambda \in \Lambda} Int_\mu Cl_\Psi(U_\lambda)$ where $Int_\mu Cl_\Psi(U_\lambda) \in \mu$ for all $\lambda \in \Lambda$ and $\Lambda$ is countable, since $X_\mu$ is $\mu$-CC, then there exists a finite subset $\Lambda_0 \subseteq \Lambda$ where $X = \bigcup_{\lambda \in \Lambda_0} Int_\mu(Cl_\Psi(U_\lambda))$. Hence, $X = \bigcup_{\lambda \in \Lambda_0} Cl_\Psi(U_\lambda))$ this shows that $X_\Psi$ is weakly $\mu$-CC.*

**Theorem 21.** *If $(X_\mu, \mathcal{H})$ is a $\mathcal{N}\mu\mathcal{H}$-CC space, then $X_\mu$ is a $\mathcal{W}\mu\mathcal{H}$-CC space.*

**Proof.** Suppose $X_\mu$ is a $\mathcal{N}\mu\mathcal{H}$-CC space. Which means that $X = \bigcup_{\lambda \in \Lambda} U_\lambda$, where $U_\lambda \in \mu$ for all $\lambda \in \Lambda$ and $\Lambda$ is a countable index set, then there exists a finite $\Lambda_0 \subseteq \Lambda$ where $X \backslash \bigcup_{\lambda \in \Lambda_0} Int_\mu Cl_\mu(U_\lambda) \in \mathcal{H}$.
However, $X \backslash \bigcup_{\lambda \in \Lambda_0} Cl_\mu(U_\lambda) \subseteq X \backslash \bigcup_{\lambda \in \Lambda_0} Int_\mu Cl_\mu(U_\lambda)$. Hence, $X \backslash \bigcup_{\lambda \in \Lambda_0} Cl_\mu(U_\lambda) \in \mathcal{H}$. □

Figure 1 shows the relationship between all types of generalization of $\mu$-compact spaces studied in this paper.

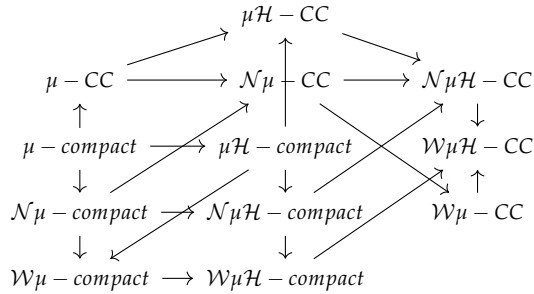

**Figure 1.** The relationship between all types of generalization of $\mu$-compact spaces.

## 4. Function Properties on $\mathcal{N}\mu$-Countably Compact and $\mathcal{W}\mu$-Countably Compact

**Theorem 22.** *Let $f : X_\mu \to Y_\beta$ be a $(\mu, \beta)$-continuous function.*

1.  *If $A$ is a $\mathcal{W}\mu$-CC subset of $X$, then $f(A)$ is $\mathcal{W}\beta$-CC.*
2.  *If $A$ is a $\mathcal{N}\mu$-CC subset of $X$, then $f(A)$ is $\mathcal{N}\beta$-CC.*

**Proof.** (1) : · Suppose $f(A) = \bigcup_{\lambda \in \Lambda} V_\lambda$, where $V_\lambda \in \beta$ for all $\lambda \in \Lambda$ and $\Lambda$ is a countable index set. Since $f$ is $(\mu, \beta)$-continuous, then $A = \bigcup_{\lambda \in \Lambda} f^{-1}(V_\lambda)$, where $f^{-1}(V_\lambda) \in \mu$ for all $\lambda \in \Lambda$ and $\Lambda$ is a countable index set and $A$ is a $\mathcal{W}\mu$-CC set. Thus, there exist $\lambda_1, \lambda_2, ..., \lambda_n \in \Lambda$ where $A \subset \bigcup_{k=1}^{n} Cl_\mu(f^{-1}(V_{\lambda_k}))$. Thus, $f(A) \subset \bigcup_{k=1}^{n} f(Cl_\mu(f^{-1}(V_{\lambda_k})))$. Since $f$ is $(\mu, \beta)$-continuous and $Cl_\mu(f^{-1}(B)) \subset f^{-1}(Cl_\beta(B))$ for all $B \subseteq Y$, then $f(Cl_\mu(f^{-1}(V_{\lambda_k}))) \subset Cl_\beta f(f^{-1}(V_{\lambda_k})) \subset Cl_\beta(V_{\lambda_k})$. Hence, $f(A)$ is $\mathcal{W}\beta$-CC.
(2) : · Suppose $f(A) = \bigcup_{\lambda \in \Lambda} V_\lambda$, where $V_\lambda \in \beta$ for all $\lambda \in \Lambda$ and $\Lambda$ is a countable index

set. Since $f$ is $(\mu, \beta)$-continuous, then $A = \bigcup_{\lambda \in \Lambda} f^{-1}(V_\lambda)$, where $f^{-1}(V_\lambda) \in \mu$ for all $\lambda \in \Lambda$ and $\Lambda$ is a countable index set and $A$ is $\mathcal{N}\mu$-CC set. Thus, there exist $\lambda_1, \lambda_2, ..., \lambda_n \in \Lambda$ where $A \subset \bigcup_{k=1}^{n} Int_\mu(Cl_\mu(f^{-1}(V_{\lambda_k})))$. Thus, $f(A) \subset \bigcup_{k=1}^{n} f(Int_\mu(Cl_\mu(f^{-1}(V_{\lambda_k}))))$. Since $f$ is $(\mu, \beta)$-continuous and $Int_\mu(Cl_\mu(f^{-1}(B))) \subset f^{-1}(Int_\beta(Cl_\beta(B)))$ for every subset $B$ of $Y$, then $f(Int_\mu(Cl_\mu(f^{-1}(V_{\lambda_k})))) \subset Int_\beta(Cl_\beta f(f^{-1}(V_{\lambda_k}))) \subset Int_\beta(Cl_\beta(V_{\lambda_k}))$. Hence, $f(A)$ is $\mathcal{N}\beta$-CC. $\square$

**Theorem 23.** *Let $f : X_\mu \to Y_\beta$ be a $(\mu, \beta)$-continuous surjective function.*

1. *If $X$ is a $\mathcal{W}\mu$-CC, then $f(X)$ is $\mathcal{W}\beta$-CC.*
2. *If $X$ is a $\mathcal{N}\mu$-CC, then $f(X)$ is $\mathcal{N}\beta$-CC.*

**Proof.** $(1) : \cdot$ Suppose $f(X) = \bigcup_{\lambda \in \Lambda} V_\lambda$, where $V_\lambda \in \beta$ for all $\lambda \in \Lambda$ and $\Lambda$ is a countable index set. Since $f$ is $(\mu, \beta)$-continuous, then $X = \bigcup_{\lambda \in \Lambda} f^{-1}(V_\lambda)$, where $f^{-1}(V_\lambda) \in \mu$ for all $\lambda \in \Lambda$ and $\Lambda$ is a countable index set and $X$ is $\mathcal{W}\mu$-CC. Thus, there exist $\lambda_1, \lambda_2, ..., \lambda_n \in \Lambda$ where $X = \bigcup_{k=1}^{n} Cl_\mu(f^{-1}(V_{\lambda_k}))$. Thus, $f(X) = \bigcup_{k=1}^{n} f(Cl_\mu(f^{-1}(V_{\lambda_k})))$. Since $f$ is $(\mu, \beta)$-continuous and $Cl_\mu(f^{-1}(B)) \subset f^{-1}(Cl_\beta(B))$ for all $B \subseteq Y$, then $f(Cl_\mu(f^{-1}(V_{\lambda_k}))) \subset Cl_\beta f(f^{-1}(V_{\lambda_k})) \subset Cl_\beta(V_{\lambda_k})$. Thus, $f(X)$ is $\mathcal{W}\beta$-CC. Hence, $Y = f(X)$ is $\mathcal{W}\beta$-CC since $f$ is surjective.

$(2) : \cdot$ Suppose $f(X) = \bigcup_{\lambda \in \Lambda} V_\lambda$, where $V_\lambda \in \beta$ for all $\lambda \in \Lambda$ and $\Lambda$ is countable index set. Since $f$ is $(\mu, \beta)$-continuous, then $X = \bigcup_{\lambda \in \Lambda} f^{-1}(V_\lambda)$, where $f^{-1}(V_\lambda) \in \mu$ for all $\lambda \in \Lambda$ and $\Lambda$ is a countable index set and $X$ is $\mathcal{W}\mu$-CC. Thus, there exist $\lambda_1, \lambda_2, ..., \lambda_n \in \Lambda$ where $X = \bigcup_{k=1}^{n} Cl_\mu(f^{-1}(V_{\lambda_k}))$. Thus, $f(X) = \bigcup_{k=1}^{n} f(Cl_\mu(f^{-1}(V_{\lambda_k})))$. Since $f$ is $(\mu, \beta)$-continuous, then $A = \bigcup_{\lambda \in \Lambda} f^{-1}(V_\lambda)$ where $f^{-1}(V_\lambda) \in \mu$ for all $\lambda \in \Lambda$ and $\Lambda$ is a countable index set and $X$ is $\mathcal{N}\mu$-CC. Thus, there exist $\lambda_1, \lambda_2, ..., \lambda_n \in \Lambda$ where $X = \bigcup_{k=1}^{n} Int_\mu(Cl_\mu(f^{-1}(V_{\lambda_k})))$. Thus, $f(X) = \bigcup_{k=1}^{n} f(Int_\mu(Cl_\mu(f^{-1}(V_{\lambda_k}))))$. Since $f$ is $(\mu, \beta)$-continuous and $Int_\mu(Cl_\mu(f^{-1}(B))) \subset f^{-1}(Int_\beta(Cl_\beta(B)))$ for all $B \subseteq Y$, then $f(Int_\mu(Cl_\mu(f^{-1}(V_{\lambda_k})))) \subset Int_\beta(Cl_\beta f(f^{-1}(V_{\lambda_k}))) \subset Int_\beta(Cl_\beta(V_{\lambda_k}))$. Thus, $f(X)$ is $\mathcal{N}\beta$-CC. Hence, $Y = f(X)$ is $\mathcal{N}\beta$-CC since $f$ is surjective. $\square$

**Theorem 24.** *Let $f : (X_\mu, \mathcal{H}) \to Y_\beta$ be a $(\mu, \beta)$-continuous surjective.*

1. *If $(X_\mu, \mathcal{H})$ is $\mathcal{W}\mu\mathcal{H}$-CC, then $Y_\beta$ is $\mathcal{W}\beta f(\mathcal{H})$-CC.*
2. *If $(X_\mu, \mathcal{H})$ is $\mathcal{N}\mu\mathcal{H}$-CC, then $Y_\beta$ is $\mathcal{N}\beta f(\mathcal{H})$-CC.*

**Proof.** $(1) : \cdot$ Suppose $f(X) = \bigcup_{\lambda \in \Lambda} V_\lambda$, where $V_\lambda \in \beta$ for all $\lambda \in \Lambda$ and $\Lambda$ is countable index set. Since $f$ is $(\mu, \beta)$-continuous, $X = \bigcup_{\lambda \in \Lambda} f^{-1}(V_\lambda)$, where $f^{-1}(V_\lambda) \in \mu$ for all $\lambda \in \Lambda$ and $\Lambda$ is a countable index and $X$ is $\mathcal{W}\mu\mathcal{H}$-CC. Thus, there exist $\lambda_1, \lambda_2, ..., \lambda_n \in \Lambda$ where $X \backslash \bigcup_{k=1}^{n} Cl_\mu(f^{-1}(V_{\lambda_k})) \in \mathcal{H}$. Since $f$ is $(\mu, \beta)$-continuous and $Cl_\mu(f^{-1}(B)) \subset f^{-1}(Cl_\beta(B))$ for all $B \subseteq Y$, then $X \backslash \bigcup_{k=1}^{n} (f^{-1}(Cl_\beta(V_{\lambda_k})) \subset X \backslash \bigcup_{k=1}^{n} Cl_\mu(f^{-1}(V_k)) \in \mathcal{H}$. Since $f(Cl_\mu(f^{-1}(V_{\lambda_k}))) \subset Cl_\beta f(f^{-1}(V_{\lambda_k})) \subset Cl_\beta(V_{\lambda_k})$. Thus, $f(X) \backslash \bigcup_{k=1}^{n} (Cl_\beta(V_{\lambda_k}) \in f(\mathcal{H})$. Since $f$ is surjective, then $f(X) = Y$. This means $Y$ is $\mathcal{W}\beta f(\mathcal{H})$-CC.

$(2) : \cdot$ Suppose $f(X) = \bigcup_{\lambda \in \Lambda} V_\lambda$, where $V_\lambda \in \beta$ for all $\lambda \in \Lambda$ and $\Lambda$ is countable index

set. Since $f$ is $(\mu, \beta)$-continuous, $X = \bigcup_{\lambda \in \Lambda} f^{-1}(V_\lambda)$, where $f^{-1}(V_\lambda) \in \mu$ for all $\lambda \in \Lambda$ and $\Lambda$ is a countable index and $X$ is $\mathcal{N}\mu\mathcal{H}$-CC. Thus, there exist $\lambda_1, \lambda_2, ..., \lambda_n \in \Lambda$ where $X \backslash \bigcup_{k=1}^{n} Int_\mu Cl_\mu(f^{-1}(V_{\lambda_k})) \in \mathcal{H}$. Since $f$ is $(\mu, \beta)$-continuous and $Int_\mu(Cl_\mu(f^{-1}(B))) \subset f^{-1}(Int_\beta(Cl_\beta(B)))$ for all $B \subseteq Y$, then

$X \backslash \bigcup_{k=1}^{n} (f^{-1}(Int_\beta(Cl_\beta(V_{\lambda_k}))) \subset X \backslash \bigcup_{k=1}^{n} Int_\mu(Cl_\mu(f^{-1}(V_k))) \in \mathcal{H}.$

Since $f(Int_\mu(Cl_\mu(f^{-1}(V_{\lambda_k})))) \subset Int_\beta(Cl_\beta f(f^{-1}(V_{\lambda_k}))) \subset Int_\beta(Cl_\beta(V_{\lambda_k})).$

Thus, $f(X) \backslash \bigcup_{k=1}^{n} Int_\beta(Cl_\beta(V_{\lambda_k}) \in f(\mathcal{H})$. Since $f$ is surjective, then $f(X) = Y$. This means $Y$ is $\mathcal{N}\beta f(\mathcal{H})$-CC. $\square$

**Theorem 25.** *Let $f : X_\mu \to (Y_\beta, \mathcal{H})$ be a $(\mu, \beta)$-open bijective function.*

1. *If $(Y_\beta, \mathcal{H})$ is $\mathcal{W}\beta\mathcal{H}$-CC, then $X_\mu$ is $\mathcal{W}\mu f^{-1}(\mathcal{H})$-CC.*
2. *If $(Y_\beta, \mathcal{H})$ is $\mathcal{N}\beta\mathcal{H}$-CC, then $X_\mu$ is $\mathcal{N}\mu f^{-1}(\mathcal{H})$-CC.*

**Proof.** Since $f : X_\mu \to (Y_\beta, \mathcal{H})$ is a $(\mu, \beta)$-open bijective, then $f^{-1} : (Y_\beta, \mathcal{H}) \to X_\mu$ is a $(\beta, \mu)$-continuous surjective. By Theorem 24, so $(Y_\beta, \mathcal{H})$ is a $\mathcal{W}\beta\mathcal{H}$-CC(resp.$\mathcal{N}\beta\mathcal{H}$-CC), then $X_\mu$ is $\mathcal{W}\mu f^{-1}(\mathcal{H})$-CC (resp.$\mathcal{N}\mu f^{-1}(\mathcal{H})$-CC). $\square$

**Theorem 26.** *Let $f : (X_\mu, \mathcal{H}) \to Y_\beta$ be a $(\mu, \beta)$-continuous.*

1. *If $A$ is $\mathcal{W}\mu\mathcal{H}$-CC, then $f(A)$ is $\mathcal{W}\beta f(\mathcal{H})$-CC.*
2. *If $A$ is $\mathcal{N}\mu\mathcal{H}$-CC, then $f(A)$ is $\mathcal{N}\beta f(\mathcal{H})$-CC.*

**Proof.** $(1) : \cdot$ Suppose $f(A) = \bigcup_{\lambda \in \Lambda} V_\lambda$, where $V_\lambda \in \beta$ for all $\lambda \in \Lambda$ and $\Lambda$ is a countable index set. Since $f$ is $(\mu, \beta)$-continuous, then $A = \bigcup_{\lambda \in \Lambda} f^{-1}(V_\lambda)$, where $f^{-1}(V_\lambda) \in \mu$ for all $\lambda \in \Lambda$ and $\Lambda$ is a countable index and $A$ is $\mathcal{W}\mu\mathcal{H}$-CC set. Thus, there exist $\lambda_1, \lambda_2, ..., \lambda_n \in \Lambda$ where $A \backslash \bigcup_{k=1}^{n} Cl_\mu(f^{-1}(V_{\lambda_k})) \in \mathcal{H}$. It is clear that $Cl_\mu(f^{-1}(V_{\lambda_k})) \subset (f^{-1}Cl_\beta(V_{\lambda_k}))$.

Thus, $A \backslash \bigcup_{k=1}^{n} (f^{-1}Cl_\beta(V_{\lambda_k})) \subset A \backslash \bigcup_{k=1}^{n} Cl_\mu(f^{-1}(V_{\lambda_k})) \in \mathcal{H}$. Thus,

$A \backslash \bigcup_{k=1}^{n} f^{-1}Cl_\beta(V_{\lambda_k}) = A \backslash \bigcup_{k=1}^{n} Cl_\beta(f^{-1}(V_{\lambda_k})) =$

$A \cap f^{-1}(Y \backslash \bigcup_{k=1}^{n} Cl_\beta(f^{-1}(V_{\lambda_k}))).$

Hence, $f(A \cap f^{-1}(Y \backslash \bigcup_{k=1}^{n} Cl_\beta(f^{-1}(V_{\lambda_k})))) = f(A) \cap (Y \backslash \bigcup_{k=1}^{n} Cl_\beta(f^{-1}(V_{\lambda_k})))$

$= f(A) \backslash \bigcup_{k=1}^{n} Cl_\beta(V_{\lambda_k}) \in f(\mathcal{H})$. This means $f(A)$ is $\mathcal{W}\beta f(\mathcal{H})$-CC.

$(2) : \cdot$ It is clear that $f$ is $(\mu, \beta)$-continuous and $Int_\mu(Cl_\mu(f^{-1}(B))) \subset f^{-1}(Int_\beta(Cl_\beta(B)))$ for all $B \subseteq Y$, then

$A \backslash \bigcup_{k=1}^{n} (f^{-1}(Int_\beta(Cl_\beta(V_{\lambda_k}))) \subset A \backslash \bigcup_{k=1}^{n} Int_\mu(Cl_\mu(f^{-1}(V_k))) \in \mathcal{H}.$

Since $f(Int_\mu(Cl_\mu(f^{-1}(V_{\lambda_k})))) \subset Int_\beta(Cl_\beta f(f^{-1}(V_{\lambda_k}))) \subset Int_\beta(Cl_\beta(V_{\lambda_k})).$

Thus $f(A) \backslash \bigcup_{k=1}^{n} Int_\beta(Cl_\beta(V_{\lambda_k}) \in f(\mathcal{H})$.This means $f(A)$ is $\mathcal{N}\beta f(\mathcal{H})$-CC. $\square$

**Theorem 27.** *Let $X_\mu$ be a $\mathcal{W}\mu$-CC; if $f : X_\mu \to Y_\beta$ is strongly $\oslash(\mu, \beta)$-continuous surjective, then $Y_\beta$ is $\beta$-CC.*

**Proof.** Suppose $Y = \bigcup_{\lambda \in \Lambda} V_\lambda$, where $V_\lambda \in \beta$ for all $\lambda \in \Lambda$ and $\Lambda$ is a countable index set. Then, for all $t \in X$, there exists $V_{\lambda_t}$ for some $\lambda_t \in \Lambda$ where $f(t) \in V_{\lambda_t}$. Since $f$ is a

strongly $\oslash(\mu, \beta)$-continuous, then $U_{\lambda_t} \in \mu$ containing $t$ such that $f(Cl_\mu(U_{\lambda_t})) \subseteq V_{\lambda_t}$. Since $\Lambda$ is countable index set, we obtain $X = \bigcup\limits_{\lambda_t \in \Lambda} U_{\lambda_t}$, where $U_{\lambda_t} \in \mu$ for all $\lambda \in \Lambda$ and $\Lambda$ is countable index set. Since $X_\mu$ is $\mathcal{W}\mu$-CC, we obtain $X = \bigcup\limits_{n=1}^{m} Cl_\mu(U_{\lambda_{t_n}})$.

Thus, $Y = f(X) = f(\bigcup\limits_{n=1}^{m} Cl_\mu(U_{\lambda_{t_n}})) = \bigcup\limits_{n=1}^{m} f(Cl_\mu(U_{\lambda_{t_n}})) \subseteq \bigcup\limits_{n=1}^{m} (V_{\lambda_{t_n}})$. Hence, $Y_\beta$ is a $\beta$-CC. $\square$

**Theorem 28.** *Let $X_\mu$ be a $\mathcal{N}\mu$-CC; if $f : X_\mu \to Y_\beta$ is super $(\mu, \beta)$-continuous surjective, then $Y_\beta$ is $\beta$-CC.*

**Proof.** Suppose $Y = \bigcup\limits_{\lambda \in \Lambda} V_\lambda$, where $V_\lambda \in \beta$ for all $\lambda \in \Lambda$ and $\Lambda$ is a countable index set. Then, for all $t \in X$, there exists $V_{\lambda_t}$ for some $\lambda_t \in \Lambda$ such that $f(t) \in V_{\lambda_t}$. Since $f$ is a super $(\mu, \beta)$-continuous, then $U_{\lambda_t} \in \mu$ containing $t$ where $f(Int_\mu Cl_\mu(U_{\lambda_t})) \subseteq V_{\lambda_t}$. Since $\Lambda$ is a countable index set, we obtain $X = \bigcup\limits_{\lambda_t \in \Lambda} U_{\lambda_t}$ where $U_{\lambda_t} \in \mu$ for all $\lambda \in \Lambda$ and $\Lambda$ is countable index set. Since $X_\mu$ is $\mathcal{N}\mu$-CC, we obtain $X = \bigcup\limits_{n=1}^{m} Int_\mu Cl_\mu(U_{\lambda_{t_n}})$.

Thus, $Y = f(X) = f(\bigcup\limits_{n=1}^{m} (Int_\mu Cl_\mu(U_{\lambda_{t_n}}))) \subseteq \bigcup\limits_{n=1}^{m} f(Int_\mu Cl_\mu(U_{\lambda_{t_n}})) \subseteq \bigcup\limits_{n=1}^{m} (V_{\lambda_{t_n}})$. Hence $Y_\beta$ is a $\beta$-CC. $\square$

**Theorem 29.** *Let $X_\mu$ be a $\mathcal{N}\mu$-CC; if $f : X_\mu \to Y_\beta$ is $(\delta, \delta')$-continuous surjective, then $Y_\beta$ is $\mathcal{N}\beta$-CC.*

**Proof.** Suppose $Y = \bigcup\limits_{\lambda \in \Lambda} Int_\beta Cl_\beta(V_\lambda)$, where $V_\lambda \in \beta$ for all $\lambda \in \Lambda$ and $\Lambda$ is a countable index set. Then, for all $t \in X$, there exists $Int_\beta Cl_\beta(V_{\lambda_t})$ for some $\lambda_t \in \Lambda$ where $f(t) \in Int_\beta Cl_\beta(V_{\lambda_t})$. Since $f$ is a $(\delta, \delta')$-continuous, then there exists $U_{\lambda_t} \in \mu$ containing $t$ where $f(Int_\mu Cl_\mu(U_{\lambda_t})) \subseteq Int_\beta Cl_\beta(V_{\lambda_t})$. Since $\Lambda$ is a countable index set, we obtain $X = \bigcup\limits_{\lambda_t \in \Lambda} U_{\lambda_t}$, where $U_{\lambda_t} \in \mu$ for all $\lambda \in \Lambda$ and $\Lambda$ is a countable index set. Since $X_\mu$ is $\mathcal{N}\mu$-CC, we obtain $X = \bigcup\limits_{n=1}^{m} Int_\mu Cl_\mu(U_{\lambda_{t_n}})$. Thus,

$Y = f(X) = f(\bigcup\limits_{n=1}^{m} Int_\mu Cl_\mu(U_{\lambda_{t_n}}) \subseteq \bigcup\limits_{n=1}^{m} f(Int_\mu Cl_\mu(U_{\lambda_{t_n}}) \subseteq \bigcup\limits_{n=1}^{m} Int_\beta Cl_\beta(V_{\lambda_{t_n}})$. Hence, $Y_\beta$ is a $\mathcal{N}\beta$-CC. $\square$

**Theorem 30.** *Let $X_\mu$ be a $\mathcal{N}\mu$-CC,*

1. *If $f : X_\mu \to Y_\beta$ is strongly $\oslash(\mu, \beta)$- continuous surjective, then $Y_\beta$ is $\beta$-CC.*
2. *If $f : X_\mu \to Y_\beta$ is super $(\mu, \beta)$- continuous surjective, then $Y_\beta$ is $\beta$-CC.*
3. *If $f : X_\mu \to Y_\beta$ is $(\delta, \delta')$- continuous surjective, then $Y_\beta$ is $\beta$-CC.*

**Proof.** It is straightforward and similar to Theorem 27, and therefore omitted. $\square$

**Theorem 31.** *Let $f : (X_\mu, \mathcal{H}) \to Y_\beta$ be almost $(\mu, \beta)$- continuous surjective.*

1. *If $(X_\mu, \mathcal{H})$ is a $\mathcal{W}\mu\mathcal{H}$-CC, then $Y_\beta$ is also $\mathcal{W}\beta f(\mathcal{H})$-CC.*
2. *If $(X_\mu, \mathcal{H})$ is a $\mathcal{N}\mu\mathcal{H}$-CC, then $Y_\beta$ is also $\mathcal{N}\beta f(\mathcal{H})$-CC.*

**Proof.** (1) : · Suppose $Y = \bigcup\limits_{\lambda \in \Lambda} V_\lambda$, where $V_\lambda \in \beta$ for all $\lambda \in \Lambda$ and $\Lambda$ is countable index set. Since $f$ is a almost $(\mu, \beta)$- continuous, then $f^{-1}(Int_\beta Cl_\beta(V_\lambda)) \in \mu$. Thus $X = \bigcup\limits_{\lambda \in \Lambda} f^{-1}(Int_\beta Cl_\beta(V_\lambda))$ for all $\lambda \in \Lambda$ is a countable index set, then there exists a finite

sub-collection $\{f^{-1}(Int_\beta Cl_\beta(V_{\lambda_k})) : k \in \mathbb{N}\}$ where $X \backslash Cl_\mu(\bigcup_{k=1}^{n} f^{-1}(Int_\beta Cl_\beta(V_{\lambda_k}))) \in \mathcal{H}$,

$$X \backslash Cl_\mu(\bigcup_{k=1}^{n} f^{-1}(Cl_\beta(\bigcup_{k=1}^{n}(V_{\lambda_k})))) \subseteq X \backslash Cl_\mu(f^{-1}(\bigcup_{k=1}^{n}(Cl_\beta(V_{\lambda_k}))))$$

$$\subseteq X \backslash Cl_\mu(\bigcup_{k=1}^{n} f^{-1}(Int_\beta Cl_\beta(V_{\lambda_k}))) \in \mathcal{H}, \text{ it is clear that}$$

$$X \backslash Cl_\mu(f^{-1}(\bigcup_{k=1}^{n}(Cl_\beta(V_{\lambda_k})))) = X \backslash (f^{-1}(\bigcup_{k=1}^{n}(Cl_\beta(V_{\lambda_k})))) \in \mathcal{H}, \text{ then}$$

$$f(X) \backslash (\bigcup_{k=1}^{n}(Cl_\beta(V_{\lambda_k}))) \in f(\mathcal{H}). \text{ Hence, } Y \text{ is a } \mathcal{W}\beta f(\mathcal{H})\text{-CC}.$$

　　(2) : · Suppose $Y = \bigcup_{\lambda \in \Lambda} V_\lambda$, where $V_\lambda \in \beta$ for all $\lambda \in \Lambda$ and $\Lambda$ is a countable index set.

Since $f$ is an almost $(\mu, \beta)$-continuous, then $f^{-1}(Int_\beta Cl_\beta(V_\lambda)) \in \mu$.

Thus, $X = \bigcup_{\lambda \in \Lambda} f^{-1}(Int_\beta Cl_\beta(V_\lambda))$ for all $\lambda \in \Lambda$ is a countable index set, then there exist

$\lambda_1, \lambda_2, ..., \lambda_n \in \Lambda$ where $X \backslash Int_\mu Cl_\mu(\bigcup_{k=1}^{n} f^{-1}(Int_\beta Cl_\beta(V_{\lambda_k}))) \in \mathcal{H}$.

Since $Int_\mu Cl_\mu(f^{-1}(V_{\lambda_k})) \subset (f^{-1}(Int_\beta Cl_\beta(V_{\lambda_k}))$, then

$$X \backslash \bigcup_{k=1}^{n} f^{-1}(Int_\beta Cl_\beta(\bigcup_{k=1}^{n}(V_{\lambda_k}))) \subseteq X \backslash Int_\mu Cl_\mu(f^{-1}(\bigcup_{k=1}^{n}(int_\beta Cl_\beta(V_{\lambda_k})))) \in \mathcal{H}.$$

Thus $X \backslash \bigcup_{k=1}^{n} f^{-1}(Int_\beta Cl_\beta(\bigcup_{k=1}^{n}(V_{\lambda_k}))) \in \mathcal{H}$, it is clear that

$$f(X \backslash (f^{-1}(\bigcup_{k=1}^{n}(Int_\beta Cl_\beta(V_{\lambda_k}))))) = f(X) \backslash (f(f^{-1}(\bigcup_{k=1}^{n}(Int_\beta Cl_\beta(V_{\lambda_k})))))$$

$$= f(X) \backslash (\bigcup_{k=1}^{n}(int_\beta Cl_\beta(V_{\lambda_k})) \in f(\mathcal{H}). \text{ Hence, } Y \text{ is a } \mathcal{N}\beta f(\mathcal{H})\text{-CC}. \quad \square$$

## 5. Applications in Soft Set Theory

　　Recall that soft set theory is an important mathematical tool in uncertainty. The concepts defined in the current paper can be applied to furnish more work to obtain generalizations of covering properties of soft generalized topological spaces. In particular, we define soft $\mu$-CC and soft $\mathcal{N}\mu$-CC as generalizations of soft $\mu$-compactness. Moreover, we provide an examined example to verify the new definitions as an applicable generalizations.

**Definition 17** ([24]). *A soft set $\mathcal{S}_\mathcal{A}$ on the universe $X$ is defined by the set of ordered pairs $\mathcal{S}_\mathcal{A} = \{(t, f_\mathcal{A}(t)) : t \in G, f_\mathcal{A}(t) \in 2^X\}$, where $\{f_\mathcal{A} : G \to 2^X\}$ and $G$ is the set of all possible parameters such that $f_\mathcal{A}(t) = \emptyset$ if $t \notin \mathcal{A}$. $\mathcal{S}_\mathcal{A}$ is said to be an approximate function of the soft set. The value of $f_\mathcal{A}(t)$ may be arbitrary. $\mathcal{S}(X)$ stands for the set of all soft sets.*

**Definition 18.** *Let $\mathcal{S}_\mathcal{A} \in \mathcal{S}(X)$.*
1.　*If $f_\mathcal{A}(t) = X$ for each $t \in G$, then $\mathcal{S}_\mathcal{A}$ is said to be an A-universal soft set, denoted by $\mathcal{S}_{\hat{\mathcal{A}}}$. If $\mathcal{A} = G$, then $\mathcal{S}_{\hat{\mathcal{A}}}$ is said to be a universal soft set, denoted by $\mathcal{S}_{\hat{G}}$ [25].*
2.　*The soft complement of $\mathcal{S}_\mathcal{A}$, denoted by $X \backslash \mathcal{S}_\mathcal{A}$, is defined by the approximate function $f_{X \backslash \mathcal{A}}(t) = X \backslash f_\mathcal{A}(t)$, where $X \backslash f_\mathcal{A}(t)$ is the complement of the set $f_\mathcal{A}(t)$ for all $t \in G$ [26].*

**Definition 19.** *Let $\mathcal{S}_\mathcal{A}, \mathcal{S}_\mathcal{B} \in \mathcal{S}(X)$.*
1.　*$\mathcal{S}_\mathcal{B}$ is a soft subset of $\mathcal{S}_\mathcal{A}$, denoted by $\mathcal{S}_\mathcal{B} \subseteq \mathcal{S}_\mathcal{A}$, if $f_\mathcal{A}(t) \subseteq f_\mathcal{B}(t)$ for all $t \in G$ [27].*
2.　*The soft union of $\mathcal{S}_\mathcal{A}$ and $\mathcal{S}_\mathcal{B}$, denoted by $\mathcal{S}_\mathcal{A} \cup \mathcal{S}_\mathcal{B}$, is defined by the approximate function $f_{\mathcal{A} \cup \mathcal{B}}(t) = f_\mathcal{A}(t) \cup f_\mathcal{B}(t)$ [25].*
3.　*The soft intersection of $\mathcal{S}_\mathcal{A}$ and $\mathcal{S}_\mathcal{B}$, denoted by $\mathcal{S}_\mathcal{A} \cap \mathcal{S}_\mathcal{B}$, is defined by the approximate function $f_{\mathcal{A} \cap \mathcal{B}}(t) = f_\mathcal{A}(t) \cap f_\mathcal{B}(t)$ [26].*

**Definition 20** ([28])**.** *Let $\mathcal{S}_\mathcal{A} \in \mathcal{S}(X)$. A soft generalized topology (briefly. sGT) on $S_A$, denoted by $\mathcal{S}_{\mathcal{A}\mu}$ is a family of soft subsets of $\mathcal{S}_\mathcal{A}$ such that $S_\varnothing \in \mu$ and if a family $\{\mathcal{S}_{\mathcal{A}_\rangle} : \mathcal{S}_{\mathcal{A}i} \subseteq \mathcal{S}_\mathcal{A}, i \in J \subseteq \mathbb{N}\} \subseteq \mu$ then $\bigcup_{i\in J}(\mathcal{S}_{\mathcal{A}i}) \in \mu$.*

**Definition 21** ([28])**.** *Let $(S_A, \mu)$ be a sGTS. Every element of $\mu$ is called a soft $\mu$-open set. The $S_\varnothing$ is a soft $\mu$-open set. If $\mathcal{S}_\mathcal{B}$ be a soft subset of $\mathcal{S}_\mathcal{A}$, then $\mathcal{S}_\mathcal{B}$ is called soft $\mu$-closed if its soft complement $X \backslash \mathcal{S}_\mathcal{B}$ is a soft $\mu$-open.*

**Definition 22** ([28])**.** *Let $(S_A, \mu)$ be a sGTS and $\mathcal{S}_\mathcal{B} \subseteq \mathcal{S}_\mathcal{A}$, then*
*(a) the soft union of all soft $\mu$-open subsets of $\mathcal{S}_\mathcal{B}$ is said to be soft $\mu$-interior of $\mathcal{S}_\mathcal{B}$ and denoted by $Int_{\mathcal{S}_\mathcal{A}\mu}\mathcal{S}_\mathcal{B}$.*
*(b) the soft intersection of all soft $\mu$-closed subsets of $\mathcal{S}_\mathcal{B}$ is said to be soft $\mu$-closure of $\mathcal{S}_\mathcal{B}$ and denoted by $Cl_{\mathcal{S}_\mathcal{A}\mu}\mathcal{S}_\mathcal{B}$.*

**Definition 23** ([29])**.** *A sGTS $(S_A, \mu)$ is called soft $\mu$-compact (denoted. soft $\mu$-C) whenever $\mathcal{S}_\mathcal{A} = \bigcup_{\lambda \in \Lambda} U_\lambda$, where $U_\lambda$ is soft $\mu$-open for all $\lambda \in \Lambda$ and $\Lambda$, then there is a finite sub-collection $\{U_\lambda : \lambda \in \Lambda_0 \subseteq \Lambda\}$ such that $\mathcal{S}_\mathcal{A} = \bigcup_{\lambda \in \Lambda_0} U_\lambda$.*

**Definition 24.** *Let $(S_A, \mu)$ be a sGTS and $\mathcal{S}_\mathcal{B} \subseteq \mathcal{S}_\mathcal{A}$, then*
1. *the soft $\mu$-regular open set if $\mathcal{S}_\mathcal{B} = Int_{\mathcal{S}_\mathcal{A}\mu}Cl_{\mathcal{S}_\mathcal{A}\mu}(\mathcal{S}_\mathcal{B})$.*
2. *the soft $\mu$-regular closed set if $\mathcal{S}_\mathcal{B} = Cl_{\mathcal{S}_\mathcal{A}\mu}Int_{\mathcal{S}_\mathcal{A}\mu}(\mathcal{S}_\mathcal{B})$.*

**Definition 25.** *A sGTS $(S_A, \mu)$ is called soft $\mu$-countably compact (denoted soft $\mu$-CC) whenever $\mathcal{S}_\mathcal{A} = \bigcup_{\lambda \in \Lambda} U_\lambda$, where $U_\lambda$ is soft $\mu$-open for all $\lambda \in \Lambda$ and $\Lambda$ countable index set, then there is a finite sub-collection $\{U_\lambda : \lambda \in \Lambda_0 \subseteq \Lambda\}$ such that $\mathcal{S}_\mathcal{A} = \bigcup_{\lambda \in \Lambda_0} U_\lambda$.*

**Definition 26.** *A sGTS $(S_A, \mu)$ is called soft nearly $\mu$-countably compact (denoted soft $\mathcal{N}\mu$-CC) whenever $\mathcal{S}_\mathcal{A} = \bigcup_{\lambda \in \Lambda} U_\lambda$, where $U_\lambda$ is soft $\mu$-open for all $\lambda \in \Lambda$ and $\Lambda$ is a countable index set, then there is a finite sub-collection $\{U_\lambda : \lambda \in \Lambda_0 \subseteq \Lambda\}$ such that $\mathcal{S}_\mathcal{A} = \bigcup_{\lambda \in \Lambda_0} Int_{\mathcal{S}_\mathcal{A}\mu}Cl_{\mathcal{S}_\mathcal{A}\mu}(U_\lambda)$.*

**Corollary 2.** *Every soft $\mu$-CC space is a soft $\mathcal{N}\mu$-CC space.*

**Proof.** It is straightforward and therefore omitted. $\square$

The converse of Corollary 2 is not true, as presented in Example 8.

**Example 8.** *Let $X = \mathbb{N}, G = \mathcal{A} = \{t_i : i \in \mathbb{N}\}$ and $\mathcal{S}_{\widehat{G}} = \{(t_i, X) : t_i \in G\}$, let $\mathcal{F} = \{(t, \{1, x\}) : x \in X, x \neq 1\}$ for each $t \in G$. Consider a sGT $\mu(\mathcal{F})$ generated on sGTS $S_{\widehat{G}}$ by the soft basis $\mathcal{F}$. Then, only $\mathcal{S}_{\widehat{G}}$ and $\mathcal{S}_\varnothing$ are soft $\mu$-regular open sets so a sGTS $(\mathcal{S}_{\widehat{G}}, \mu(\mathcal{F}))$ is soft $\mathcal{N}\mu(\mathcal{F})$-CC, but it is not soft $\mu(\mathcal{F})$-CC, since a family $\left\{\mathcal{S}_{\widehat{G}_i} : i \in \mathbb{N}\right\}$, where*
$$\mathcal{S}_{\widehat{G}_1} = \{(t_1, \{1, 2\}), (t_2, \{1, 2, 3\}), (t_3, \{1, 2, 3, 4\}), \ldots \ldots\},$$
$$\mathcal{S}_{\widehat{G}_2} = \{(t_1, \{1, 3\}), (t_2, \{1, 2, 4\}), (t_3, \{1, 2, 3, 5\}), \ldots \ldots\},$$
$$\mathcal{S}_{\widehat{G}_3} = \{(t_1, \{1, 4\}), (t_2, \{1, 2, 5\}), (t_3, \{1, 2, 3, 6\}), \ldots \ldots\}$$

$$\vdots$$

*is soft $\mu(\mathcal{F})$-open cover of sGTS $(\mathcal{S}_{\widehat{G}}, \mu(\mathcal{F}))$ with no finite soft $\mu(\mathcal{F})$-open sub-cover.*

## 6. Conclusions

We have explored and examined the definition of weakly (nearly) $\mu$-countably compact spaces in the sense of generalized topology given in [1]. Further, we studied the effect of hereditary classes on these spaces. The space presented in Example 1 is $\mathcal{N}\mu$-CC, but not $\mu$-CC. Some other results regarding subsets of such spaces have been presented. Observing

that $\mu$- countably compactness is a generalization of $\mu$-compactness, Figure 1 is a summary to show the relations between these spaces studied in the paper and other spaces generalizing $\mu$-compactness. Finally, we studied the effect of generalized continuity on these spaces. In particular, it is proved that the images and preimages of the new notions of spaces defined in this paper are preserved under $(\mu, \beta)$-continuous functions. Stronger results are given if we use strongly $\varnothing(\mu, \beta)$-continuous functions and super $(\mu, \beta)$-continuous functions. More varying results are given by using $(\delta, \delta')$-continuous functions and almost $(\mu, \beta)$-continuous functions.

As future research, some modifications can be made if we replace the generalized topology $\mu$ by a weaker framework as a weaker structure $\mathcal{WS}$ [30]. Moreover, we can study the effect of soft $\mu$-regular sets on soft nearly $\mu$-countably compact spaces defined in Section 5. To see some applications of generalizations of spaces in generalized topology, you can see [29,31,32].

**Author Contributions:** Conceptualization, Z.A. and A.B.; investigation, Z.A. and E.A.-Z., writing, review, and editing; A.B. and I.J. All authors have read and agreed to the published version of the manuscript.

**Funding:** This research received no external funding.

**Data Availability Statement:** Not applicable.

**Acknowledgments:** The authors would like to thank the referees for their useful comments, and suggestions.

**Conflicts of Interest:** The authors declare no conflict of interest.

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
