# Peer review of "Weakly and Nearly Countably Compactness in Generalized Topology"

_axioms, doi:10.3390/axioms12020122_

Round 1
Author Response
Yes, I am attaching the reply.

Reviewer 2 Report
The paper, after minor revision according to the attached report, deserve to be publishd. Best regards.

Author Response
YEes, we replied the second report

Reviewer 3 Report
The authors introduced the notions of weakly µ-countably compact space, and nearly µ-countably compact space. We study the effect of hereditary class on weakly µ-countably compact space and nearly µ-countably compact space. Moreover, we use properties of functions to investigate the effects of some types of continuity on weakly µ-countably compact spaces and nearly µ-countably compact spaces. Furthermore, differences between these two new spaces and the other known spaces are investigated.
The following are the specific comments that I suggest the authors to revise:
There are many typos errors.
Abstract should be revised in a scientific view.
The similarity of the paper is seems to be high.
In section 4, the line 394 “Y is Wβf(H)-CC” instead of “Y is Wβ(H)-CC”.
In section 4, the line 403 “Y is Nβf(H)-CC” instead of “Y is Nβ(H)-CC”.
Although authors provided many examples, they are not practical examples. It is recommended that authors concretize the parameters and universe in the examples to improve readability.
The authors only presented a mathematical framework, but there is no specific application to illustrate applicable.
Author Response
Yes, we reply the third report.

Round 2
Reviewer 1 Report
Dear the authores,
I have checked your revised paper. All are OK.
Reviewer 3 Report
Can be accepted, now.